# Rethinking the Power of Timestamps for Robust Time Series Forecasting: A Global-Local Fusion Perspective

**Chengsen Wang**[1*] **Qi Qi**[1*] **Jingyu Wang**[12†]
**Haifeng Sun**[1] **Zirui Zhuang**[1] **Jinming Wu**[1] **Jianxin Liao**[1]

[1]Beijing University of Posts and Telecommunications, Beijing, China
[2]Pengcheng Laboratory, Shenzhen, China
{cswang, qiqi8266, wangjingyu}@bupt.edu.cn
{hfsun, zhuangzirui, wjm_18, liaojx}@bupt.edu.cn

## Abstract

Time series forecasting has played a pivotal role across various industries, including finance, transportation, energy, healthcare, and climate. Due to the abundant seasonal information they contain, timestamps possess the potential to offer robust global guidance for forecasting techniques. However, existing works primarily focus on local observations, with timestamps being treated merely as an optional supplement that remains underutilized. When data gathered from the real world is polluted, the absence of global information will damage the robust prediction capability of these algorithms. To address these problems, we propose a novel framework named GLAFF. Within this framework, the timestamps are modeled individually to capture the global dependencies. Working as a plugin, GLAFF adaptively adjusts the combined weights for global and local information, enabling seamless collaboration with any time series forecasting backbone. Extensive experiments conducted on nine real-world datasets demonstrate that GLAFF significantly enhances the average performance of widely used mainstream forecasting models by 12.5%, surpassing the previous state-of-the-art method by 5.5%. Code is available at https://github.com/ForestsKing/GLAFF.

## 1 Introduction

Time series forecasting holds significant importance across various industries, including finance [1, 12], transportation [4, 11], energy [28, 31], healthcare[16, 29], and climate [8, 44]. With the development of deep learning techniques, neural network-based methods [15, 18, 40, 47] have notably propelled advancements owing to their strong capability in capturing dependencies within time series. The relevant models have evolved from statistical models to RNNs, CNNs, Transformers, and LLMs. However, existing research primarily concentrates on local observations within history sliding windows, overlooking the significance of timestamps.

Due to the abundant seasonal information they contain, timestamps possess the potential to offer robust global guidance for forecasting techniques. For instance, traffic volumes on weekdays typically exhibit high peaks. Regrettably, existing works primarily focus on local observations, with timestamps being treated merely as an optional supplement that remains underutilized. DLinear [41] and FPT [47] completely overlook timestamps. Informer [45] and TimesNet [38] incorporate timestamps by summing their embeddings with position embeddings and data embeddings. These intertwined

---

*Equal contribution.
†Corresponding author.

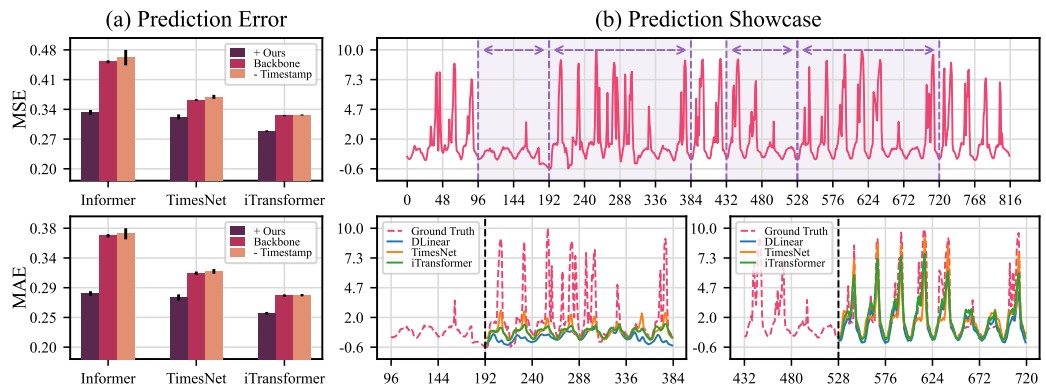

Figure 1: The experimental results on Traffic dataset. (a) illustrates the outcomes of the ablation study on mainstream forecasting models and their variants. (b) depicts the visualization of traffic volume (upper), successful prediction case (lower right), and failed prediction case (lower left), respectively.

patterns encourage networks to extract information from more intuitive observations. iTransformer [24] embeds timestamp features separately into tokens employed by the attention mechanism. This embedding method across time points damaged the physical significance of timestamps. To validate this proposition, we conduct an ablation study on the aforementioned models using the Traffic dataset. The results depicted in Figure 1(a) indicate that the performance of the models exhibits no significant decline after removing timestamps. Meanwhile, our proposed GLAFF demonstrates a notable enhancement in mainstream forecasting models.

Moreover, Time series collected from the real world often be polluted [5]. For example, a spike in electricity consumption coupled with short circuits can induce point anomalies, while a reduction in traffic volume coinciding with holidays can evoke contextual anomalies. When local information gathered from the real world contains anomalies, the absence of global information will damage the robust prediction capability of most forecasting techniques [7, 39, 43, 46]. We illustrate traffic volume for San Francisco Bay area freeways at an hourly granularity in Figure 1(b). Typically, this sequence exhibits a clear periodic pattern, alternating with five high peaks (weekdays) and two low peaks (weekends). However, due to a holiday, the week from 24 to 192 shows a deviation, resulting in three high peaks and four low peaks. As evident from the illustration in the lower right corner of Figure 1(b), the mainstream forecasting models [24, 38, 41] usually demonstrate reliable prediction capability. Nonetheless, when the observations within the history window include anomalies, as illustrated in the lower left corner of Figure 1(b), these models are significantly affected and yield notably underestimated predictions. Therefore, it is necessary to reasonably incorporate more robust global information into the existing forecasting technique.

To address the aforementioned problems, we propose a generalized framework named GLAFF (short for **G**lobal-**L**ocal **A**daptive **F**usion **F**ramework), aimed at enhancing the robust prediction capability of time series forecasting models in the real world leveraging global information. Specifically, GLAFF initially employs the Attention-based Mapper to individually model the timestamps containing global information and maps them to observations conforming to a standard distribution. Subsequently, to handle scenarios where anomalies are present within the observations of the sliding window, we utilize the Robust Denormalizer to inverse normalize the initial mappings, thereby mitigating the impact of data drift [17]. Finally, the Adaptive Combiner dynamically adjusts the combined weights for global mapping and local prediction within the prediction window, yielding the final prediction outcome. By fusing the robustness of global information with the flexibility of local information, GLAFF demonstrates a substantial enhancement in the robust prediction capability of mainstream forecasting models. Additionally, GLAFF serves as a model-agnostic and plug-and-play framework that can seamlessly collaborate with any time series forecasting backbone.

In general, the contributions of our paper are summarised as follows:

- We propose GLAFF that leverages global information, represented by timestamps, to improve the robust prediction capability of time series forecasting models. GLAFF is a plug-and-play module that seamlessly collaborates with any time series forecasting backbone.

- We design a Robust Denormalization module to facilitate the adaptation of GLAFF for data drift, even when the observations encompass anomalies, alongside an Adaptive Combiner module for dynamically fusing global and local information.

- We conduct comprehensive experiments on nine real-world benchmark datasets across five domains. The result demonstrates that GLAFF significantly improves the robust prediction capability of mainstream forecasting models.

## 2  Related Work

As a significant real-world challenge, time series forecasting has garnered considerable attention. Initially, ARIMA [2] establishes an autoregressive model and performs forecasts in a moving average manner. However, the inherent complexity of the real world often renders such statistical methodologies [2, 14, 33] challenging to adapt. With the development of deep learning techniques, neural network-based methods have become increasingly important. Recurrent neural networks [10, 13, 30] dynamically capture temporal dependencies by modeling semantic information within a sequential structure. Unfortunately, this architecture suffers from gradient vanishing/exploding and information forgetting when dealing with long sequences. To further improve prediction performance, self-attention mechanisms [19, 22, 45] and convolutional networks [21, 36, 38] have been introduced to capture long-range dependencies. Additionally, prior research [41] has demonstrated that a simple linear network augmented by decomposition can also achieve competitive performance. Nowadays, with fast growth and remarkable performances of large language models, there is a growing interest [3, 32, 47] in utilizing LLM to analyze time series data. Recently, the iTransformer [24] has emerged as the state-of-the-art method for time series forecasting tasks by embedding series from different channels into the variate tokens utilized by the attention mechanism.

Most time series forecasting techniques focus on local observations, with timestamps being treated merely as an optional supplement that remains underutilized. DLinear [41], FPT [47], and other models [26, 40, 43] completely overlook timestamps. When data gathered from the real world is polluted, the absence of global information will damage the robust prediction capability of these algorithms. Informer [45], TimesNet [38], and other models [23, 37, 46] incorporate timestamps by summing their embeddings with position embeddings and data embeddings. These intertwined patterns encourage networks to extract information from more intuitive observations. iTransformer [24] embeds timestamp features separately into tokens employed by the attention mechanism. This embedding method across time points damaged the physical significance of timestamps.

The processing of timestamps by the previous baselines and our proposed GLAFF can be abstracted as early fusion (feature-level fusion) and late fusion (decision-level fusion). Early fusion integrates modalities into a single representation at the input level and processes the fused representation through the model. Late fusion allows each modality to run independently through its own model and fuses the outputs of each modality. Compared to early fusion, late fusion maximizes the processing effectiveness of each modality and is less susceptible to the noise of a single modality, resulting in greater robustness and reliability. This has been validated by extensive previous work [20, 27, 35].

## 3  Methodology

We propose a model-agnostic and plug-and-play framework, GLAFF, which utilizes global information, represented by timestamps, to enhance the robust prediction capability of mainstream time series forecasting models in real-world scenarios. In multivariate time series forecasting, given the history observations of $c$ channels within $h$ time steps $\mathbf{X} = \{\mathbf{x}_1, \ldots, \mathbf{x}_h\} \in \mathbb{R}^{h \times c}$, we aim to forecast the subsequent $p$ time steps $\mathbf{Y} = \{\mathbf{x}_{h+1}, \ldots, \mathbf{x}_{h+p}\} \in \mathbb{R}^{p \times c}$. In addition to observations, we incorporate timestamps to provide global information. For each timestamp, we extract its month, day, weekday, hour, minute, and second as timestamp features, respectively. For instance, for the timestamp *2018-06-02 12:00:00* at moment $t$, its feature representation is $\mathbf{s}_t = [06, 02, 05, 12, 00, 00] \in \mathbb{R}^{1 \times 6}$. The holiday markers may also be included if accessible. Unlike observations, the timestamp features within the history window $\mathbf{S} = \{\mathbf{s}_1, \ldots, \mathbf{s}_h\} \in \mathbb{R}^{h \times 6}$ and the timestamp features within the prediction window $\mathbf{T} = \{\mathbf{s}_{h+1}, \ldots, \mathbf{s}_{h+p}\} \in \mathbb{R}^{p \times 6}$ are known. In this section, we describe the detailed workflow of the entire GLAFF framework and explain how it fuses local information $\mathbf{X}$ and global information $\mathbf{S}, \mathbf{T}$ to predict $\mathbf{Y}$.

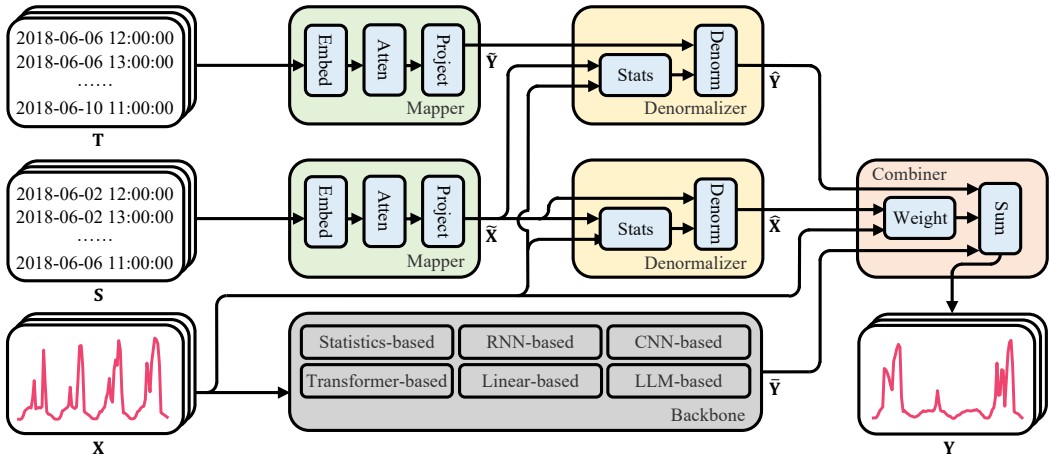

Figure 2: The overall architecture of GLAFF mainly consists of three primary components: Attention-based Mapper, Robust Denormalizer, and Adaptive Combiner.

## 3.1 Overview

GLAFF is a plug-and-play framework that seamlessly collaborates with any time series forecasting backbone. The overall architecture of the plugin is depicted in Figure 2, comprising three primary components: Attention-based Mapper, Robust Denormalizer, and Adaptive Combiner. Following local prediction $\bar{\mathbf{Y}} \in \mathbb{R}^{p \times c}$ provided by the backbone network based on history observations $\mathbf{X}$ (maybe including underutilized history timestamps $\mathbf{S}$ and future timestamps $\mathbf{T}$), GLAFF leverages global information to revise it. Initially, the Attention-based Mapper captures dependencies between timestamps through an attention mechanism, mapping timestamp features $\mathbf{S}$ and $\mathbf{T}$ into an initial history mapping $\tilde{\mathbf{X}} \in \mathbb{R}^{h \times c}$ and an initial future mapping $\tilde{\mathbf{Y}} \in \mathbb{R}^{p \times c}$, conforming to standard distribution. Subsequently, the Robust Denormalizer inverse normalizes the initial mappings $\tilde{\mathbf{X}}$ and $\tilde{\mathbf{Y}}$ to $\hat{\mathbf{X}} \in \mathbb{R}^{h \times c}$ and $\hat{\mathbf{Y}} \in \mathbb{R}^{p \times c}$ based on quantile deviation between the initial mapping $\tilde{\mathbf{X}}$ and the actual observations $\mathbf{X}$ within the history window, mitigating the impact of data drift. Lastly, the Adaptive Combiner dynamically adjusts the combined weights of the global mapping $\hat{\mathbf{Y}}$ and the local prediction $\bar{\mathbf{Y}}$ within the prediction window according to the disparity between the final mapping $\hat{\mathbf{X}}$ and the actual observations $\mathbf{X}$ within the history window, yielding the final prediction outcome $\mathbf{Y}$. By fusing the robustness of global information and the flexibility of local information, GLAFF significantly enhances the robust prediction capability of mainstream forecasting models.

## 3.2 Attention-based Mapper

As depicted in the green segment of Figure 2, our proposed Attention-based Mapper employs a simplified encoder-only architecture within the Transformer [34] framework, comprising an embedding layer, attention blocks, and a projection layer. Analogous to typical Transformer-based encoders [37, 45], each timestamp feature is initially tokenized by an embedding layer to describe its properties, applied by self-attention for mutual interactions, and individually processed by feed-forward networks for series representations. Subsequently, a projection layer is utilized to acquire the initial mappings. Leveraging the capability of the attention mechanism for capturing long-range dependencies and parallel computation, the Attention-based Mapper can sufficiently model the global information embodied by timestamps.

Specifically, in Attention-based Mapper, the procedure for obtaining its corresponding initial mapping $\tilde{\mathbf{X}}$, which conforms to the standard distribution, based on the history timestamps $\mathbf{S}$, is succinctly delineated as follows:

$$
\begin{aligned}
\mathbf{H}^0 &= \text{Embedding}\left(\mathbf{S}\right) \\
\mathbf{H}^{i+1} &= \text{Attention}\left(\mathbf{H}^i\right), \ i = 0, \cdots, l-1 \\
\tilde{\mathbf{X}} &= \text{Projection}\left(\mathbf{H}^l\right)
\end{aligned}
\tag{1}
$$

where $\mathbf{H}^i \in \mathbb{R}^{h \times d}$ denotes the intermediate feature variable output from the $i$-th attention block, and $d$ represents the dimension of the intermediate feature variable. The attention blocks are stacked with $l$ layers to capture the high-level semantic information hidden within the timestamps. To maintain simplicity in implementation, both the embedding and projection layers are comprised of a single linear layer. Following conventional protocol, the primary computation steps for the $i$-th attention block are outlined as:

$$\mathbf{H}^i = \text{LayerNorm}\left(\mathbf{H}^i + \text{MSA}\left(\mathbf{H}^i, \mathbf{H}^i, \mathbf{H}^i\right)\right)$$
$$\mathbf{H}^{i+1} = \text{LayerNorm}\left(\mathbf{H}^i + \text{FeedForward}\left(\mathbf{H}^i\right)\right) \tag{2}$$

where $\text{LayerNorm}\left(\cdot\right)$ represents the commonly adopted layer normalization and $\text{FeedForward}\left(\cdot\right)$ denotes the multilayer feedforward network. The $\text{MSA}\left(\mathbf{Q}, \mathbf{K}, \mathbf{V}\right)$ indicates the Multihead Self-Attention mechanism [34], where $\mathbf{Q}, \mathbf{K}, \mathbf{V}$ serve as the query, key, and value respectively. Additionally, a dropout mechanism is incorporated to alleviate overfitting and enhance the generalization of the network. The process of obtaining the corresponding initial mapping $\tilde{\mathbf{Y}}$ based on the future timestamps $\mathbf{T}$, conforming to the standard distribution, mirrors the aforementioned procedure, simply substituting $\mathbf{S}$ and $\tilde{\mathbf{X}}$ in Equation 1 with $\mathbf{T}$ and $\tilde{\mathbf{Y}}$ respectively.

## 3.3 Robust Denormalizer

Due to the inherent variability of the real world, time series observations typically undergo rapid evolution over time, a phenomenon commonly referred to as data drift [17]. This phenomenon can result in discrepancies across different time spans and hinder the generalization ability of deep learning models. Recognizing the presence of data drift, GLAFF employs a two-phase untangling modeling strategy to address the global information represented by timestamps. In the first phase, the network in Attention-based Mapper produces initial mappings, denoted as $\tilde{\mathbf{X}}$ and $\tilde{\mathbf{Y}}$, which are assumed to satisfy a standard distribution for reducing the difficulty of modeling the dependencies between timestamps and observations. Subsequently, in the second phase, leveraging the distribution deviations between the initial mapping $\tilde{\mathbf{X}}$ and the actual observations $\mathbf{X}$ within the history window, the Robust Denormalizer separately inverse normalizes the initial mappings $\tilde{\mathbf{X}}$ and $\tilde{\mathbf{Y}}$ to produce the final mappings $\hat{\mathbf{X}}$ and $\hat{\mathbf{Y}}$, mitigating the impact of data drift.

To alleviate the impact of data drift, a feasible solution [9, 17, 23, 25] has been proposed: removing dynamic factors from the original data through a normalization procedure before feeding them into the deep learning model, and subsequently reintroducing these dynamic factors via an inverse normalization procedure after output from the deep learning model. The conventional inverse normalization procedure typically considers distribution deviations in mean and standard deviation. Nonetheless, this approach is susceptible to extreme values and lacks robustness when the observations contain anomalies. Instead of relying on mean and standard deviation, we employ median and quantile ranges [6], respectively, to enhance the robustness of the Robust Denormalizer against anomalies. As depicted in the yellow segment of Figure 2, the procedure for Robust Denormalizer to inverse normalize initial mappings $\tilde{\mathbf{X}}$ and $\tilde{\mathbf{Y}}$ into final mappings $\hat{\mathbf{X}}$ and $\hat{\mathbf{Y}}$ can be succinctly expressed as:

$$\hat{\mathbf{X}} = \frac{\tilde{\mathbf{X}} - \tilde{\mu}}{\tilde{\sigma}} \times \sigma + \mu$$
$$\hat{\mathbf{Y}} = \frac{\tilde{\mathbf{Y}} - \tilde{\mu}}{\tilde{\sigma}} \times \sigma + \mu \tag{3}$$

where $\tilde{\mu} \in \mathbb{R}^{1 \times c}$ and $\mu \in \mathbb{R}^{1 \times c}$ represent the median of the initial mapping $\tilde{\mathbf{X}}$ and the actual observation $\mathbf{X}$ for each channel, respectively. Similarly, $\tilde{\sigma} \in \mathbb{R}^{1 \times c}$ and $\sigma \in \mathbb{R}^{1 \times c}$ denote the quantile range (the distance between the $q$ quantile and the $1 - q$ quantile) of the initial mapping $\tilde{\mathbf{X}}$ and the actual observation $\mathbf{X}$ for each channel. Specifically, when $q = 0.75$, $\tilde{\sigma}$ and $\sigma$ correspond to the inter-quartile range (IQR[3]) for each channel of the initial mapping $\tilde{\mathbf{X}}$ and the actual observation $\mathbf{X}$.

## 3.4 Adaptive Combiner

Owing to the intricacies of real-world scenarios, data preferences for model bias will continuously change with online concept drifts [42]. Therefore, we need a data-dependent strategy to change

---

[3]IQR is defined as the difference between the 1st and 3rd quartiles of a distribution or set of values, and is a robust measure of the distribution spread.

the model selection policy continuously. In other words, the combined weights of global and local information necessitate adaptive and dynamic updates. When the time series pattern exhibits clarity and stability, greater emphasis should be placed on robust global information. Conversely, increased attention should be directed towards flexible local information when the time series pattern appears ambiguous and variable. Within the framework of GLAFF, we employ an Adaptive Combiner to realize the adaptive adjustment of combined weights.

As illustrated in the red segment of Figure 2, the Adaptive Combiner initially dynamically adjusts the combined weights of the global mapping $\hat{\mathbf{Y}}$ and the local prediction $\bar{\mathbf{Y}}$ within the prediction window, based on the deviation between the final mapping $\hat{\mathbf{X}}$ and the actual observation $\mathbf{X}$ within the history window. Subsequently, we aggregate the dual-source information based on the combined weights to yield the final prediction $\mathbf{Y}$. Specifically, the primary computation procedure of the Adaptive Combiner is represented as:

$$\mathbf{W} = \mathrm{MLP}\left(\hat{\mathbf{X}} - \mathbf{X}\right)$$
$$\mathbf{Y} = \sum \mathbf{W} \times \left(\hat{\mathbf{Y}} \oplus \bar{\mathbf{Y}}\right)$$

(4)

where $\mathbf{W} \in \mathbb{R}^{1 \times c \times 2}$ signifies the dynamically generated combined weight by the network based on the deviation between the final mapping $\hat{\mathbf{X}}$ and the actual observation $\mathbf{X}$ within the history window. The $\oplus$ denotes the concatenation operation based on the additional last dimension, and $\sum$ denotes the summation operation performed across the last dimension. For simplicity, the weight generation network consists solely of a Multilayer Perceptron (MLP) containing a hidden layer and a layer of $\mathrm{Softmax}$ for weight normalization.

Through adaptive adjustment of combined weights, our method can effectively fuses the robustness of global information and the flexibility of local information, thereby enhancing its suitability for intricate and fluctuating real-world scenarios.

## 4 Experiment

### 4.1 Experimental Setup

**Dataset**    We conduct extensive experiments on nine datasets across five domains, including Electricity, Exchange, Traffic, Weather, and ILI, along with four ETT datasets. Detailed dataset information is provided in Appendix A.1. We follow the standard segmentation protocol [24, 37, 45], strictly dividing each dataset into training, validation, and testing sets chronologically to ensure no information leakage issues. The segmentation ratio for each dataset is set to 6:2:2. Regarding prediction settings, we also adhere to established mainstream protocols [26, 38, 41]. Specifically, we set the length of the history window to 96 for the Electricity, Exchange, Traffic, Weather, and four ETT datasets, while the prediction length varies within {96, 192, 336, 720}. For the ILI, which has fewer time points, the length of the history window is fixed at 36, and the prediction length varies within {24, 36, 48, 60}.

**Backbone**    To demonstrate the effectiveness of the framework, we select several mainstream forecasting models based on different architectures, including the Transformer-based Informer (2021) [45] and iTransformer (2024) [24], the Linear-based DLinear (2023) [41], and the Convolution-based TimesNet (2023) [38]. Notably, iTransformer represents the previous state-of-the-art method in time series forecasting tasks. Further details regarding the backbone models are provided in Appendix A.2. As described in Section 2, these backbones encompass three different treatments for timestamps employed in prior forecasting techniques, namely summation (Informer, TimesNet), concatenation (iTransformer), and omission (DLinear).

The details of experimental setup can be found in Appendix A.3. All experiments are based on our runs, utilizing the same hardware configurations, and repeated 3 times with different random seeds.

### 4.2 Main Result

Table 1 compares the prediction outcomes for mainstream baselines and GLAFF. We present a detailed version of this table in Appendix B.1. The results indicate that GLAFF significantly surpasses all four widely used mainstream baselines across all nine real-world benchmark datasets. In particular, GLAFF enhances the respective backbones by an average of 12.5%.

Table 1: The forecasting errors for multivariate time series among GLAFF and mainstream baselines. A lower outcome indicates a better prediction. The best results are highlighted in bold.

| | Method | Informer | | + Ours | | DLinear | | + Ours | | TimesNet | | + Ours | | iTransformer | | + Ours | | Impr. |
|---|---|---|---|---|---|---|---|---|---|---|---|---|---|---|---|---|---|---|
| | | MSE | MAE | MSE | MAE | MSE | MAE | MSE | MAE | MSE | MAE | MSE | MAE | MSE | MAE | MSE | MAE | |
| Electricity | 96 | 0.333 | 0.414 | **0.217** | **0.323** | 0.196 | 0.283 | **0.147** | **0.238** | 0.175 | 0.280 | **0.154** | **0.248** | 0.153 | 0.246 | **0.120** | **0.198** | 15.7% |
| | 192 | 0.362 | 0.444 | **0.220** | **0.329** | 0.196 | 0.286 | **0.172** | **0.253** | 0.191 | 0.293 | **0.169** | **0.261** | 0.167 | 0.259 | **0.143** | **0.216** | |
| | 336 | 0.352 | 0.434 | **0.230** | **0.337** | 0.208 | 0.301 | **0.197** | **0.274** | 0.211 | 0.310 | **0.185** | **0.276** | 0.183 | 0.276 | **0.168** | **0.240** | |
| | 720 | 0.364 | 0.443 | **0.247** | **0.351** | 0.239 | 0.331 | **0.239** | **0.308** | 0.235 | 0.326 | **0.226** | **0.303** | 0.220 | 0.310 | **0.217** | **0.279** | |
| ETTh1 | 96 | 0.926 | 0.736 | **0.609** | **0.569** | 0.409 | 0.440 | **0.391** | **0.418** | 0.453 | 0.481 | **0.435** | **0.464** | 0.420 | 0.454 | **0.411** | **0.441** | 8.8% |
| | 192 | 1.235 | 0.844 | **0.831** | **0.680** | 0.457 | 0.475 | **0.446** | **0.457** | 0.533 | 0.531 | **0.520** | **0.517** | 0.494 | 0.502 | **0.474** | **0.482** | |
| | 336 | 1.354 | 0.875 | **0.882** | **0.698** | 0.500 | 0.506 | **0.492** | **0.488** | 0.621 | 0.580 | **0.596** | **0.560** | 0.538 | 0.528 | **0.534** | **0.519** | |
| | 720 | 1.264 | 0.857 | **0.937** | **0.730** | 0.610 | 0.576 | **0.609** | **0.556** | 0.844 | 0.697 | **0.773** | **0.661** | 0.716 | 0.629 | **0.704** | **0.615** | |
| ETTh2 | 96 | 0.708 | 0.549 | **0.422** | **0.443** | 0.159 | 0.278 | **0.128** | **0.205** | 0.183 | 0.298 | **0.174** | **0.276** | 0.177 | 0.287 | **0.172** | **0.271** | 12.1% |
| | 192 | 1.133 | 0.688 | **0.860** | **0.599** | 0.187 | 0.309 | **0.165** | **0.238** | 0.218 | 0.329 | **0.204** | **0.306** | 0.199 | 0.311 | **0.196** | **0.295** | |
| | 336 | 0.997 | 0.667 | **0.747** | **0.570** | 0.207 | 0.330 | **0.206** | **0.265** | 0.240 | 0.346 | **0.219** | **0.319** | 0.220 | 0.329 | **0.213** | **0.306** | |
| | 720 | 1.607 | 0.815 | **1.255** | **0.720** | 0.262 | 0.378 | **0.214** | **0.288** | 0.281 | 0.376 | **0.278** | **0.363** | 0.271 | 0.366 | **0.270** | **0.356** | |
| ETTm1 | 96 | 0.593 | 0.548 | **0.503** | **0.486** | 0.339 | 0.388 | **0.309** | **0.353** | 0.449 | 0.448 | **0.381** | **0.398** | 0.383 | 0.415 | **0.349** | **0.386** | 8.1% |
| | 192 | 0.611 | 0.576 | **0.534** | **0.525** | 0.394 | 0.418 | **0.362** | **0.386** | 0.448 | 0.461 | **0.440** | **0.432** | 0.429 | 0.445 | **0.403** | **0.420** | |
| | 336 | 0.888 | 0.726 | **0.707** | **0.628** | 0.450 | 0.451 | **0.442** | **0.432** | 0.550 | 0.504 | **0.494** | **0.462** | 0.485 | 0.479 | **0.468** | **0.460** | |
| | 720 | 1.037 | 0.786 | **0.925** | **0.719** | 0.508 | 0.493 | **0.493** | **0.467** | 0.619 | 0.559 | **0.563** | **0.507** | 0.566 | 0.532 | **0.564** | **0.519** | |
| ETTm2 | 96 | 0.186 | 0.311 | **0.147** | **0.266** | 0.115 | 0.232 | **0.080** | **0.165** | 0.121 | 0.234 | **0.110** | **0.212** | 0.120 | 0.235 | **0.111** | **0.221** | 12.1% |
| | 192 | 0.242 | 0.348 | **0.230** | **0.341** | 0.143 | 0.261 | **0.109** | **0.193** | 0.155 | 0.267 | **0.136** | **0.239** | 0.149 | 0.266 | **0.144** | **0.252** | |
| | 336 | 0.454 | 0.466 | **0.308** | **0.380** | 0.176 | 0.294 | **0.148** | **0.229** | 0.190 | 0.293 | **0.175** | **0.269** | 0.185 | 0.293 | **0.182** | **0.283** | |
| | 720 | 0.861 | 0.616 | **0.719** | **0.561** | 0.225 | 0.340 | **0.221** | **0.274** | 0.242 | 0.334 | **0.225** | **0.309** | 0.233 | 0.333 | **0.232** | **0.327** | |
| Exchange | 96 | 0.735 | 0.728 | **0.223** | **0.391** | 0.051 | 0.164 | **0.046** | **0.155** | 0.076 | 0.198 | **0.066** | **0.177** | 0.058 | 0.172 | **0.051** | **0.158** | 16.4% |
| | 192 | 1.016 | 0.861 | **0.421** | **0.547** | 0.099 | 0.238 | **0.093** | **0.225** | 0.135 | 0.272 | **0.115** | **0.242** | 0.113 | 0.245 | **0.108** | **0.231** | |
| | 336 | 1.331 | 0.971 | **0.691** | **0.694** | 0.174 | 0.317 | **0.161** | **0.299** | 0.237 | 0.363 | **0.219** | **0.336** | 0.210 | 0.339 | **0.196** | **0.314** | |
| | 720 | 2.054 | 1.263 | **1.152** | **0.922** | 0.314 | 0.446 | **0.308** | **0.439** | 0.636 | 0.618 | **0.595** | **0.582** | 0.517 | 0.551 | **0.510** | **0.529** | |
| ILI | 24 | 3.374 | 1.356 | **2.487** | **1.106** | 2.087 | 1.131 | **1.875** | **0.963** | 1.478 | 0.713 | **1.333** | **0.684** | 1.148 | 0.659 | **1.129** | **0.658** | 9.9% |
| | 36 | 3.094 | 1.293 | **2.617** | **1.157** | 2.065 | 1.107 | **1.756** | **0.957** | 1.294 | 0.748 | **1.204** | **0.695** | 1.061 | 0.695 | **1.039** | **0.682** | |
| | 48 | 3.383 | 1.370 | **2.879** | **1.230** | 2.059 | 1.088 | **1.639** | **0.912** | 1.280 | 0.736 | **1.278** | **0.721** | 1.209 | 0.735 | **1.164** | **0.715** | |
| | 60 | 3.610 | 1.415 | **3.086** | **1.274** | 2.186 | 1.097 | **1.644** | **0.889** | 1.291 | 0.773 | **1.191** | **0.713** | 1.222 | 0.758 | **1.196** | **0.731** | |
| Traffic | 96 | 0.467 | 0.375 | **0.352** | **0.297** | 0.482 | 0.378 | **0.301** | **0.260** | 0.360 | 0.314 | **0.322** | **0.278** | 0.308 | 0.272 | **0.283** | **0.249** | 19.5% |
| | 192 | 0.455 | 0.371 | **0.343** | **0.288** | 0.449 | 0.356 | **0.302** | **0.261** | 0.364 | 0.314 | **0.325** | **0.277** | 0.327 | 0.279 | **0.291** | **0.253** | |
| | 336 | 0.462 | 0.378 | **0.335** | **0.281** | 0.453 | 0.358 | **0.306** | **0.263** | 0.373 | 0.320 | **0.331** | **0.282** | 0.338 | 0.285 | **0.301** | **0.259** | |
| | 720 | 0.495 | 0.400 | **0.340** | **0.287** | 0.475 | 0.374 | **0.327** | **0.278** | 0.396 | 0.337 | **0.339** | **0.289** | 0.357 | 0.302 | **0.320** | **0.273** | |
| Weather | 96 | 1.422 | 0.867 | **0.642** | **0.554** | 0.198 | 0.258 | **0.176** | **0.244** | 0.188 | 0.238 | **0.171** | **0.226** | 0.178 | 0.223 | **0.159** | **0.220** | 9.6% |
| | 192 | 1.429 | 0.880 | **0.877** | **0.664** | 0.237 | 0.296 | **0.219** | **0.280** | 0.234 | 0.278 | **0.233** | **0.277** | 0.231 | 0.268 | **0.214** | **0.265** | |
| | 336 | 1.796 | 1.008 | **1.506** | **0.851** | 0.282 | 0.333 | **0.265** | **0.312** | 0.293 | 0.317 | **0.288** | **0.314** | 0.289 | 0.310 | **0.273** | **0.306** | |
| | 720 | 1.542 | 0.946 | **1.427** | **0.853** | 0.343 | 0.379 | **0.330** | **0.360** | 0.368 | 0.365 | **0.364** | **0.362** | 0.370 | 0.363 | **0.352** | **0.355** | |
| Impr. | | 23.8% | | | | 13.1% | | | | 7.5% | | | | 5.5% | | | | 12.5% |

By fusing the robustness of global information with the flexibility of local information, GLAFF can significantly improve the robust prediction capability of mainstream forecasting models. Specifically, in the case of DLinear, a Linear-based model that entirely disregards timestamps, GLAFF enhances its prediction accuracy by 13.1%. For Transformer-based Informer and Convolution-based TimesNet utilizing simple timestamp summation, GLAFF yields performance improvements of 23.8% and 7.5%, respectively. In the case of Transformer-based iTransformer employing direct timestamp concatenation, GLAFF still produces a 5.5% improvement in accuracy. Additionally, we note a diminishing boosting effect of GLAFF as the modeling prowess of the backbone increases, indicative of the complementary nature of global and local information. Nonetheless, for the state-of-the-art iTransformer, GLAFF continues to offer substantial benefits.

It is evident that the enhancement of GLAFF varies across datasets with different characteristics. For datasets such as Traffic and Electricity, characterized by a significant number of channels and clear periodic patterns, GLAFF demonstrates superior capability in capturing the dependencies between timestamps and observations, resulting in performance enhancements of 19.5% and 15.7%, respectively. For the non-stationary datasets [25], such as ETTh2, ETTm2, and Exchange, the Robust Denormalizer effectively alleviates the impact of data drift, thereby augmenting prediction accuracy by 12.1%, 12.1%, and 16.4%, respectively. Regarding common datasets like ETTh1, ETTm1, Weather, and ILI, although the performance of GLAFF may not be as remarkable, it still yields improvements of 8.8%, 8.1%, 9.6%, and 9.6%, respectively.

Given the burgeoning interest in leveraging LLMs for time series, we assess the performance augmentation of GLAFF when applied to LLM-based backbones. Specifically, we employ the widely recognized FPT (2023) [47] as our baseline. FPT completely disregards timestamps similar to DLinear. We deploy two structures, GPT2(3) and GPT2(6), of FPT as outlined in their paper. The ILI dataset has a history window length of 36 and a prediction window length of 48, while other datasets have a history window length of 96 and a prediction window length of 192. As delineated in the findings presented in Table 2, GLAFF also offer significant benefits to the LLM-based baselines.

Table 2: The forecasting errors for multivariate time series among GLAFF and LLM-based baselines. A lower outcome indicates a better prediction. The best results are highlighted in bold.

| Method | GPT2(3) | | +Ours | | GPT2(6) | | +Ours | |
|---|---|---|---|---|---|---|---|---|
| | MSE | MAE | MSE | MAE | MSE | MAE | MSE | MAE |
| Electricity | 0.194±0.002 | 0.278±0.002 | **0.168±0.005** | **0.258±0.002** | 0.194±0.001 | 0.279±0.002 | **0.171±0.002** | **0.258±0.001** |
| ETTh1 | 0.466±0.001 | 0.483±0.001 | **0.454±0.003** | **0.462±0.003** | 0.468±0.002 | 0.483±0.002 | **0.445±0.002** | **0.451±0.002** |
| ETTh2 | 0.190±0.000 | 0.304±0.000 | **0.178±0.017** | **0.284±0.010** | 0.190±0.002 | 0.303±0.002 | **0.177±0.005** | **0.286±0.003** |
| ETTm1 | 0.411±0.005 | 0.431±0.003 | **0.388±0.009** | **0.409±0.005** | 0.412±0.002 | 0.431±0.002 | **0.390±0.012** | **0.413±0.007** |
| ETTm2 | 0.142±0.001 | 0.257±0.001 | **0.120±0.001** | **0.235±0.002** | 0.142±0.002 | 0.256±0.002 | **0.119±0.002** | **0.233±0.002** |
| Exchange | 0.110±0.001 | 0.238±0.002 | **0.090±0.002** | **0.219±0.002** | 0.106±0.001 | 0.234±0.001 | **0.088±0.001** | **0.216±0.001** |
| ILI | 1.585±0.041 | 0.900±0.019 | **1.393±0.024** | **0.782±0.023** | 1.494±0.012 | 0.854±0.010 | **1.396±0.026** | **0.778±0.018** |
| Traffic | 0.370±0.002 | 0.309±0.002 | **0.296±0.002** | **0.256±0.001** | 0.371±0.002 | 0.312±0.001 | **0.301±0.003** | **0.262±0.002** |
| Weather | 0.241±0.000 | 0.276±0.000 | **0.234±0.004** | **0.268±0.003** | 0.243±0.001 | 0.278±0.001 | **0.228±0.002** | **0.264±0.002** |

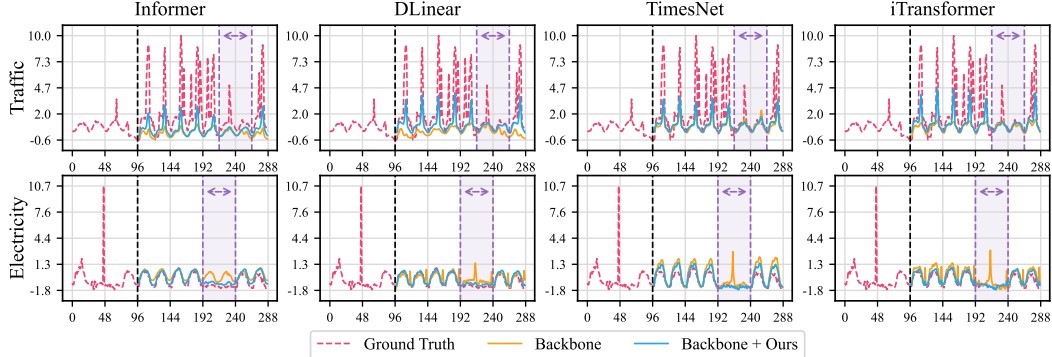

Figure 3: The illustration of prediction showcases among GLAFF and mainstream baselines.

Additionally, to validate the practical applicability of GLAFF, we assess the computation costs in Appendix B.4. The results indicate that GLAFF has little effect on model training and deployment across most scenarios, particularly when considering its significant accuracy enhancement.

## 4.3 Prediction Showcase

In addition to evaluation metrics, forecasting quality is crucial. To further compare GLAFF and the four mainstream forecasting models, we illustrate prediction showcases for two representative datasets in Figure 3. We provide the full prediction showcases for the nine datasets in Appendix B.5. It is evident that GLAFF can yield more realistically robust predictions. At the same time, the respective backbones are susceptible to abnormal local information.

The Traffic dataset records traffic volume in hourly granularity. Typically, this sequence exhibits a clear periodic pattern, alternating with five high peaks (weekdays) and two low peaks (weekends). However, owing to a holiday, the initial two days within the example history window do not exhibit the high peaks as usual. Due to the limited local information containing such contextual anomalies, Informer, DLinear, and iTransformer all think the prediction window should also consist of only low peaks. Although TimesNet generates predictions with high peaks, it displays an incorrect alternation between five high peaks and one low peak. By introducing sufficiently modeled global information, GLAFF has enabled the four mainstream forecasting backbones to recognize the existence of high peaks and the correct periodic patterns, thus yielding more accurate forecasts.

The Electricity dataset records electricity consumption in hourly granularity. Typically, this sequence exhibits a clear periodic pattern, alternating with five peaks (weekdays) and two flat segments (weekends). However, owing to a short circuit, the middle two days within the example history window show a spike in electricity consumption. Due to the limited local information containing such point anomalies, DLinear, TimesNet, and iTransformer all think the flat segments in the prediction window should also contain a spike. Although Informer generates predictions without spikes, it completely ignores the presence of flat segments. By introducing sufficiently modeled global information, GLAFF has enabled the four mainstream forecasting backbones to recognize the contingency of spikes and the correct periodic patterns, thus yielding more robust predictions.

Table 3: The forecasting errors for multivariate time series of ablation study among GLAFF and variants. A lower outcome indicates a better prediction. The best results are highlighted in bold.

| Method | | iTransformer | | + Ours | | w/o Backbone | | w/o Attention | | w/o Quantile | | w/o Adaptive | |
|---|---|---|---|---|---|---|---|---|---|---|---|---|---|
| | | MSE | MAE | MSE | MAE | MSE | MAE | MSE | MAE | MSE | MAE | MSE | MAE |
| Electricity | 96 | 0.1525 | 0.2460 | **0.1197** | **0.1979** | 0.2058 | 0.2663 | 0.1518 | 0.2450 | 0.1467 | 0.2465 | 0.1574 | 0.2502 |
| | 192 | 0.1674 | 0.2593 | **0.1434** | **0.2157** | 0.2097 | 0.2793 | 0.1684 | 0.2610 | 0.1677 | 0.2662 | 0.1740 | 0.2662 |
| | 336 | 0.1830 | 0.2762 | **0.1683** | **0.2395** | 0.2454 | 0.3014 | 0.1832 | 0.2775 | 0.1993 | 0.2954 | 0.1953 | 0.2877 |
| | 720 | 0.2199 | 0.3097 | **0.2169** | **0.2786** | 0.2984 | 0.3386 | 0.2182 | 0.3092 | 0.2593 | 0.3403 | 0.2330 | 0.3171 |
| Traffic | 96 | 0.3084 | 0.2717 | **0.2828** | **0.2485** | 0.3348 | 0.2723 | 0.3172 | 0.2806 | 0.2909 | 0.2684 | 0.2930 | 0.2612 |
| | 192 | 0.3267 | 0.2794 | **0.2909** | **0.2528** | 0.3387 | 0.2736 | 0.3357 | 0.2884 | 0.2948 | 0.2737 | 0.2970 | 0.2610 |
| | 336 | 0.3381 | 0.2850 | **0.3005** | **0.2594** | 0.3460 | 0.2794 | 0.3482 | 0.2958 | 0.3023 | 0.2804 | 0.3082 | 0.2706 |
| | 720 | 0.3574 | 0.3015 | **0.3201** | **0.2730** | 0.3558 | 0.2906 | 0.3684 | 0.3113 | 0.3249 | 0.2984 | 0.3212 | 0.2819 |
| Weather | 96 | 0.1784 | 0.2229 | **0.1587** | **0.2199** | 0.2382 | 0.2695 | 0.1780 | 0.2214 | 0.1811 | 0.2270 | 0.1914 | 0.2379 |
| | 192 | 0.2308 | 0.2675 | **0.2138** | **0.2654** | 0.2882 | 0.3105 | 0.2383 | 0.2733 | 0.2364 | 0.2768 | 0.2489 | 0.2832 |
| | 336 | 0.2892 | 0.3099 | **0.2733** | **0.3058** | 0.3381 | 0.3414 | 0.2932 | 0.3146 | 0.2905 | 0.3134 | 0.3070 | 0.3251 |
| | 720 | 0.3701 | 0.3634 | **0.3520** | **0.3547** | 0.4011 | 0.3813 | 0.3752 | 0.3664 | 0.3727 | 0.3649 | 0.3829 | 0.3722 |
| Avg. | | 0.2602 | 0.2827 | **0.2367** | **0.2593** | 0.3000 | 0.3004 | 0.2646 | 0.2870 | 0.2555 | 0.2876 | 0.2591 | 0.2845 |

## 4.4 Ablation Study

We provide a comprehensive ablation study to validate the necessity of the GLAFF components. We implement our approach and its four variants on the iTransformer backbone. The results of our experiments on three representative benchmark datasets are presented in Table 3. Detailed results for the nine real-world benchmark datasets are available in Appendix B.2.

In w/o Backbone, we completely remove the backbone network within the GLAFF and map the future using only timestamps. Surprisingly, GLAFF still demonstrates favorable prediction performance without any observations. The average prediction accuracy of GLAFF even outperforms Informer and DLinear, and is also competitive with TimesNet and iTransformer. Global information proves adequate in scenarios featuring clear periodic and stable distributions.

In w/o Attention, we substitute the stacked attention blocks with MLP networks having the equivalent size. Following the replacement of attention blocks, GLAFF fails to capture the dependencies among timestamps adequately. It proves challenging to map out precise observations solely from a single timestamp. Particularly notable is the most marked decline in performance on the Traffic dataset, which has the largest number of channels, indicating the greatest modeling challenge for GLAFF.

In w/o Quantile, we replace the Robust Denormalizer with conventional inverse normalization. The experimental results illustrate that our design yields enhancements across all three datasets, particularly in the Electricity dataset. When the history window encompasses anomalies, conventional inverse normalization yields inaccurate estimates for distribution. Leveraging more robust quantiles, our Robust Denormalizer demonstrates enhanced robustness in mitigating the impacts of data drift.

In w/o Adaptive, we substitute the Adaptive Combiner with a straightforward averaging for global mapping and local prediction. We distinctly find that dynamically adjusting the combined weights can prove more efficacious in accommodating fluctuating real-world scenarios, particularly evident in the non-stationary Weather dataset. By fusing global and local information adaptively, GLAFF can seamlessly collaborate with any time series forecasting backbone.

## 5 Conclusion

In this work, our focus lies in leveraging global information, as denoted by timestamps, to enhance the robust prediction capability of time series forecasting models in the real world. We introduce a new approach named GLAFF, serving as a model-agnostic and plug-and-play framework. Within this framework, the timestamps are modeled individually to capture the global dependencies. Through adaptive adjustment of combined weights for global and local information, GLAFF facilitates seamless collaboration with any time series forecasting backbone. To substantiate the superiority of our approach, we have conducted comprehensive experiments on widely used benchmark datasets, demonstrating the substantial enhancement GLAFF provides to mainstream forecasting models. We hope that GLAFF can be used as a foundational component for time series forecasting and call on the community to give more attention to global information represented by timestamps.

## Acknowledgments and Disclosure of Funding

This work was supported by the National Natural Science Foundation of China under Grants (U23B2001, 62171057, 62101064, 62201072, 62001054, 62071067), the Ministry of Education and China Mobile Joint Fund (MCM20200202, MCM20180101), Beijing University of Posts and Telecommunications-China Mobile Research Institute Joint Innovation Center, China Postdoctoral Science Foundation (2023TQ0039), National Postdoctoral Program for Innovative Talents under Grant BX20230052, and the BUPT Excellent Ph.D. Students Foundation (CX20241016).

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

# A    Detailed Experimental Setup

## A.1    Dataset

We conduct extensive experiments on nine real-world datasets acorss five domains, including Electricity, Exchange, Traffic, Weather, and ILI, along with four ETT datasets. Table 4 summarizes the statistics of these datasets. These datasets have been widely utilized for benchmarking purposes and are publicly available. (1) **Electricity**[4] comprises hourly electricity consumption for 321 customers from 2012 to 2014. (2) **Exchange**[5] encompasses panel data on daily exchange rates for 8 countries from 1990 to 2019. (3) **Traffic**[6] aggregates hourly road occupancy rates measured by 862 sensors on San Francisco Bay Area freeways from 2015 to 2016. (4) **Weather**[7] captures 21 weather parameters monitored every 10 minutes from Germany in 2020. (5) **ILI**[8] records the percentage of patients with influenza-like illness and the total number of such patients collected weekly by the United States Centers for Disease Control and Prevention from 2002 to 2020. (6) **ETT**[9] records the oil temperature and load characteristics of two power transformers from 2016 to 2018, each at 2 different resolutions (15 minutes and 1 hour), resulting in a total of four datasets: ETTm1, ETTm2, ETTh1, and ETTh2.

Table 4: The statistics of each dataset. Channel represents the variate number of each dataset. Length indicates the total number of time points. Frequency denotes the sampling interval of time points.

| Dataset | Channel | Length | Frequency | Information |
|---|---|---|---|---|
| Electricity | 321 | 26304 | 1 Hour | Energy |
| Exchange | 8 | 7588 | 1 Day | Finance |
| Traffic | 862 | 17544 | 1 Hour | Transportation |
| Weather | 21 | 52696 | 10 Minutes | Climate |
| ILI | 7 | 966 | 1 Week | Healthcare |
| ETTh1 & ETTh2 | 7 | 17420 | 1 Hour | Energy |
| ETTm1 & ETTm2 | 7 | 69680 | 15 Minutes | Energy |

## A.2    Backbone

To demonstrate the effectiveness of our framework, we select several mainstream forecasting models based on different architectures, including the Transformer-based Informer (2021) and iTransformer (2024), the Linear-based DLinear (2023), and the Convolution-based TimesNet (2023). All aforementioned models are non-autoregressive forecasting models. (1) **Informer**[10] utilizes the ProbSparse attention and distillation mechanism to manage exceedingly long input sequences efficiently and incorporates a generative decoder to mitigate the error accumulation inherent in autoregressive forecasting methodologies. (2) **DLinear**[11] employs decomposition-enhanced simple linear networks to attain competitive forecasting performance. (3) **TimesNet**[12] accurately models two-dimensional dependencies by transforming the one-dimensional time series into a collection of two-dimensional tensors, leveraging multiple periods to embed intra-periodic and inter-periodic variations along the columns and rows of the tensor, respectively. (4) **iTransformer**[13] embeds individual channel into token employed by the attention mechanism, facilitating the capture of inter-channel multivariate correlations while applying a feed-forward network to each token to acquire nonlinear representations.

All backbones are based on our runs, using the same hardware. We utilize official or open-source implementations and follow the hyperparameter configurations recommended in their papers.

---

[4]https://archive.ics.uci.edu/dataset/321/electricityloaddiagrams20112014

[5]https://github.com/laiguokun/multivariate-time-series-data

[6]https://pems.dot.ca.gov

[7]https://www.bgc-jena.mpg.de/wetter

[8]https://gis.cdc.gov/grasp/fluview/fluportaldashboard.html

[9]https://github.com/zhouhaoyi/ETDataset

[10]https://github.com/zhouhaoyi/Informer2020

[11]https://github.com/cure-lab/LTSF-Linear

[12]https://github.com/thuml/TimesNet

[13]https://github.com/thuml/iTransformer

**Algorithm 1** GLAFF

```python
import torch
from torch import nn

class GLAFF(nn.Module):
    def __init__(self, hist_len, channel, dim=512, dff=2048, dropout=0.1, head_num=8, layer_num=2):
        """
        :param hist_len: the length of the history window
        :param channel: the number of the dataset channel
        :param dim: the dimension of the MultiHeadAttention
        :param dff: the dimension of the feedforward network
        :param dropout: the dropout proportion of the MultiHeadAttention
        :param head_num: the number of the attention head in the MultiHeadAttention
        :param layer_num: the number of the attention block in the Attention-based Mapper
        """

        super(GLAFF, self).__init__()

        self.Mapper = nn.Sequential(
            nn.Linear(6, dim),
            nn.TransformerEncoder(
                nn.TransformerEncoderLayer(
                    d_model=dim,
                    nhead=head_num,
                    dim_feedforward=dff,
                    dropout=dropout,
                    activation='gelu',
                    batch_first=True,
                ),
                num_layers=layer_num,
                norm=nn.LayerNorm(dim)
            ),
            nn.Linear(dim, channel)
        )

        self.Combiner = nn.Sequential(
            nn.Linear(hist_len, dff),
            nn.GELU(),
            nn.Linear(dff, 2),
            nn.Softmax(dim=-1)
        )

    def forward(self, hist_gt, hist_ts, pred_pr, pred_ts, q=0.75):
        """
        :param hist_gt: the true value in the history window
        :param hist_ts: the timestamps in the history window
        :param pred_pr: the prediction value of the backbone in the prediction window
        :param pred_ts: the timestamps in the prediction window
        :param q: the quantile of the Robust Denormalizer
        """

        # map
        hist_map = self.Mapper(hist_ts)  # the mapping value of the mapper in the history window
        pred_map = self.Mapper(pred_ts)  # the mapping value of the mapper in the prediction window

        # inverse normalize
        means_gt = torch.median(hist_gt, 1, True)[0]
        means_map = torch.median(hist_map, 1, True)[0]
        stdev_gt = torch.quantile(hist_gt, q, 1, True) - torch.quantile(hist_gt, 1 - q, 1, True)
        stdev_map = torch.quantile(hist_map, q, 1, True) - torch.quantile(hist_map, 1 - q, 1, True)
        hist_map = (hist_map - means_map) / stdev_map * stdev_gt + means_gt
        pred_map = (pred_map - means_map) / stdev_map * stdev_gt + means_gt

        # combine
        error = hist_gt - hist_map
        weight = self.Combiner(error.permute(0, 2, 1)).unsqueeze(1)
        pred = torch.stack([pred_map, pred_pr], dim=-1)
        pred = torch.sum(pred * weight, dim=-1)

        return pred
```

## A.3  Implementation

We employ the Adam optimizer and $L_2$ loss for model optimization, initializing the learning rate at $10^{-4}$. The batch size is uniformly set to 32, and the number of training epochs is fixed at 10. Optimal hyperparameters for GLAFF are determined through grid search, employing a common setup shared across all datasets and backbones. The number $l$ of the attention blocks in the Attention-based Mapper is designated as 2 (selected from {1,2,3,4,5}), and while proportion $p$ of dropout is set to 0.1 (selected from {0.0,0.1,0.2,0.3,0.4}). The quantile $q$ in the Robust Denormalizer is configured to 0.75, selected from {0.55, 0.65, 0.75, 0.85, 0.95}. We provide details of each hyperparameter in

Appendix B.3. All experiments are conducted using Python 3.10.13 and PyTorch 2.1.2, executed on an Ubuntu server equipped with an AMD Ryzen 9 7950X 16-Core processor and a single NVIDIA GeForce RTX 4090 graphics card, with each experiment repeated three times using different random seeds. If not explicitly stated, we report the Mean Square Error (MSE) and Mean Absolute Error (MAE) as evaluation metrics, with lower values indicating superior performance.

The detailed implementation of GLAFF is delineated in Algorithm 1. For simplification, the Adaptive Combiner is implemented through a straightforward two-layer linear network. We employ the *nn.TransformerEncoder()* within PyTorch to realize the Attention-based Mapper. The source code and checkpoints have been made openly accessible to facilitate future research.

# B  Full Experimental Result

## B.1  Robustness Analysis

We report in Table 5 the means and standard deviations of the evaluation metrics for the baselines and GLAFF under three runs using different random seeds, facilitating the assessment of their robustness in long-range and ultra-long-range time series forecasting tasks. Evident from the mean of the experimental outcomes, GLAFF consistently demonstrates marked superiority over all four mainstream forecasting models across all nine real-world benchmark datasets. The standard deviation indicates the consistent stability and robustness of our proposed framework.

## B.2  Ablation Study

We provide a comprehensive ablation study to validate the necessity of the GLAFF components. We implement our approach and its four variants on the iTransformer backbone. Specifically, in w/o Backbone, we completely remove the backbone network within the GLAFF and map the future using only timestamps. In w/o Attention, we substitute the stacked attention blocks with MLP networks having the equivalent size. In w/o Quantile, we replace the Robust Denormalizer with conventional inverse normalization. In w/o Adaptive, we substitute the Adaptive Combiner with a straightforward averaging mechanism for global mapping and local prediction. Due to space limitations, Section 4.4 presents experimental results for only three representative datasets, so we provide the complete results for nine real-world datasets in also Table 6.

When excluding any observations within the history window, GLAFF demonstrates an average prediction accuracy superior to Informer across nine real-world datasets and remains competitive with DLinear. However, in contrast to the three representative datasets outlined in Section 4.4, the average prediction accuracy of GLAFF across the nine datasets still lags behind TimesNet and iTransformer. We find that this discrepancy primarily stems from the highly non-stationary ILI dataset. In highly non-stationary scenarios, relying solely on global information fails to adapt to the intricate and fluctuating nature of real-world conditions. Our experimental outcomes validate the indispensable nature of both the robustness of global information and the flexibility of local information for accurate time series prediction.

The experimental results show that the adaptive adjustment of combined weights within the Adaptive Combiner is the most important among various designs. When the time series pattern exhibits clarity and stability, greater emphasis should be placed on robust global information. Conversely, increased attention should be directed towards flexible local information when the time series pattern appears ambiguous and variable. By fusing global and local information adaptively, GLAFF can seamlessly collaborate with any time series forecasting backbone to adapt to intricate and fluctuating real-world scenarios. Furthermore, the stacked attention blocks within the Attention-based Mapper and the robust inverse normalization within the Robust Denormalizer also yield notable contributions to enhancing the forecasting performance of GLAFF through efficient modeling of timestamps and robust mitigation of data drift.

It is noteworthy that the lack of either component design will corrupt the outcome of the GLAFF, resulting in a prediction accuracy lower than the standard iTransformer baseline. This aligns with both common knowledge and theoretical expectations. In terms of global information, the introduction of insufficient (w/o Attention), inaccurate (w/o Quantile), or crude (w/o Adaptive) will all hurt GLAFF. This again validates that each component design in GLAFF is reasonable and necessary.

Table 5: The complete forecasting errors for multivariate time series among GLAFF and mainstream baselines. A lower outcome indicates a better prediction. The best results are highlighted in bold.

| Method | | Informer | | + Ours | | DLinear | | + Ours | |
|---|---|---|---|---|---|---|---|---|---|
| | | MSE | MAE | MSE | MAE | MSE | MAE | MSE | MAE |
| Electricity | 96 | 0.3328±0.0053 | 0.4138±0.0007 | **0.2173±0.0007** | **0.3232±0.0007** | 0.1962±0.0001 | 0.2830±0.0005 | **0.1470±0.0008** | **0.2378±0.0011** |
| | 192 | 0.3623±0.0274 | 0.4444±0.0244 | **0.2201±0.0027** | **0.3292±0.0034** | 0.1962±0.0001 | 0.2861±0.0003 | **0.1715±0.0039** | **0.2530±0.0010** |
| | 336 | 0.3519±0.0026 | 0.4342±0.0014 | **0.2302±0.0015** | **0.3374±0.0024** | 0.2080±0.0000 | 0.3008±0.0003 | **0.1965±0.0114** | **0.2736±0.0050** |
| | 720 | 0.3640±0.0179 | 0.4427±0.0147 | **0.2472±0.0017** | **0.3507±0.0003** | 0.2392±0.0002 | 0.3305±0.0005 | **0.2390±0.0076** | **0.3082±0.0036** |
| ETTh1 | 96 | 0.9255±0.0694 | 0.7361±0.0367 | **0.6088±0.0157** | **0.5688±0.0071** | 0.4085±0.0004 | 0.4399±0.0002 | **0.3909±0.0099** | **0.4181±0.0097** |
| | 192 | 1.2354±0.0679 | 0.8437±0.0146 | **0.8312±0.0976** | **0.6797±0.0474** | 0.4572±0.0006 | 0.4752±0.0003 | **0.4462±0.0108** | **0.4565±0.0083** |
| | 336 | 1.3541±0.0747 | 0.8754±0.0107 | **0.8816±0.0249** | **0.6978±0.0033** | 0.5002±0.0005 | 0.5064±0.0011 | **0.4922±0.0040** | **0.4879±0.0021** |
| | 720 | 1.2641±0.0292 | 0.8572±0.0101 | **0.9366±0.0367** | **0.7301±0.0199** | 0.6098±0.0012 | 0.5757±0.0001 | **0.6088±0.0028** | **0.5560±0.0011** |
| ETTh2 | 96 | 0.7083±0.0800 | 0.5493±0.0301 | **0.4219±0.0393** | **0.4427±0.0186** | 0.1589±0.0006 | 0.2776±0.0013 | **0.1282±0.0051** | **0.2049±0.0055** |
| | 192 | 1.1329±0.1858 | 0.6879±0.0493 | **0.8599±0.0622** | **0.5994±0.0343** | 0.1869±0.0025 | 0.3091±0.0028 | **0.1651±0.0135** | **0.2382±0.0068** |
| | 336 | 0.9972±0.1608 | 0.6666±0.0601 | **0.7471±0.1531** | **0.5702±0.0558** | 0.2066±0.0021 | 0.3304±0.0025 | **0.2064±0.0249** | **0.2650±0.0077** |
| | 720 | 1.6066±0.3017 | 0.8152±0.0768 | **1.2550±0.0489** | **0.7203±0.0270** | 0.2621±0.0026 | 0.3783±0.0021 | **0.2135±0.0079** | **0.2875±0.0064** |
| ETTm1 | 96 | 0.5932±0.0320 | 0.5475±0.0126 | **0.5031±0.0060** | **0.4862±0.0026** | 0.3385±0.0003 | 0.3877±0.0006 | **0.3085±0.0230** | **0.3529±0.0130** |
| | 192 | 0.6111±0.0117 | 0.5763±0.0038 | **0.5339±0.0191** | **0.5248±0.0074** | 0.3936±0.0007 | 0.4179±0.0007 | **0.3619±0.0071** | **0.3857±0.0058** |
| | 336 | 0.8881±0.0376 | 0.7257±0.0206 | **0.7066±0.0932** | **0.6278±0.0387** | 0.4502±0.0009 | 0.4514±0.0004 | **0.4416±0.0208** | **0.4320±0.0090** |
| | 720 | 1.0374±0.0480 | 0.7859±0.0104 | **0.9254±0.0230** | **0.7185±0.0099** | 0.5075±0.0010 | 0.4932±0.0010 | **0.4930±0.0068** | **0.4669±0.0026** |
| ETTm2 | 96 | 0.1860±0.0089 | 0.3109±0.0103 | **0.1473±0.0303** | **0.2656±0.0279** | 0.1148±0.0003 | 0.2318±0.0012 | **0.0803±0.0010** | **0.1649±0.0020** |
| | 192 | 0.2421±0.0286 | 0.3481±0.0229 | **0.2300±0.0457** | **0.3405±0.0351** | 0.1425±0.0002 | 0.2605±0.0005 | **0.1085±0.0024** | **0.1932±0.0036** |
| | 336 | 0.4542±0.0614 | 0.4664±0.0287 | **0.3082±0.0022** | **0.3798±0.0076** | 0.1760±0.0012 | 0.2939±0.0019 | **0.1477±0.0027** | **0.2292±0.0046** |
| | 720 | 0.8609±0.2400 | 0.6156±0.0665 | **0.7192±0.0489** | **0.5611±0.0154** | 0.2252±0.0010 | 0.3396±0.0011 | **0.2211±0.0483** | **0.2737±0.0080** |
| Exchange | 96 | 0.7349±0.0501 | 0.7282±0.0226 | **0.2226±0.0059** | **0.3905±0.0042** | 0.0505±0.0005 | 0.1642±0.0013 | **0.0462±0.0004** | **0.1552±0.0007** |
| | 192 | 1.0158±0.0258 | 0.8607±0.0146 | **0.4214±0.0237** | **0.5471±0.0147** | 0.0988±0.0006 | 0.2375±0.0013 | **0.0929±0.0015** | **0.2247±0.0022** |
| | 336 | 1.3307±0.0715 | 0.9708±0.0177 | **0.6912±0.0292** | **0.6939±0.0150** | 0.1739±0.0038 | 0.3170±0.0039 | **0.1612±0.0028** | **0.2993±0.0038** |
| | 720 | 2.0536±0.0518 | 1.2631±0.0132 | **1.1516±0.0718** | **0.9222±0.0254** | 0.3137±0.0018 | 0.4460±0.0004 | **0.3077±0.0019** | **0.4386±0.0039** |
| ILI | 24 | 3.3735±0.1341 | 1.3561±0.0583 | **2.4865±0.1512** | **1.1055±0.0405** | 2.0871±0.0224 | 1.1314±0.0037 | **1.8745±0.0541** | **0.9629±0.0193** |
| | 36 | 3.0944±0.0858 | 1.2927±0.0099 | **2.6173±0.1699** | **1.1570±0.0638** | 2.0654±0.0323 | 1.1074±0.0077 | **1.7557±0.0834** | **0.9571±0.0201** |
| | 48 | 3.3828±0.0680 | 1.3701±0.0308 | **2.8792±0.0996** | **1.2296±0.0415** | 2.0586±0.0133 | 1.0879±0.0040 | **1.6393±0.0430** | **0.9115±0.0166** |
| | 60 | 3.6102±0.1638 | 1.4147±0.0313 | **3.0858±0.1401** | **1.2743±0.0478** | 2.1861±0.0212 | 1.0969±0.0039 | **1.6441±0.0333** | **0.8894±0.0124** |
| Traffic | 96 | 0.4668±0.0265 | 0.3748±0.0150 | **0.3521±0.0125** | **0.2969±0.0087** | 0.4821±0.0001 | 0.3782±0.0001 | **0.3007±0.0006** | **0.2602±0.0011** |
| | 192 | 0.4549±0.0043 | 0.3713±0.0031 | **0.3430±0.0094** | **0.2882±0.0056** | 0.4487±0.0001 | 0.3561±0.0003 | **0.3019±0.0036** | **0.2609±0.0021** |
| | 336 | 0.4622±0.0088 | 0.3778±0.0054 | **0.3352±0.0009** | **0.2812±0.0023** | 0.4526±0.0001 | 0.3577±0.0002 | **0.3062±0.0011** | **0.2631±0.0002** |
| | 720 | 0.4945±0.0106 | 0.3997±0.0073 | **0.3403±0.0166** | **0.2870±0.0101** | 0.4749±0.0001 | 0.3735±0.0002 | **0.3267±0.0028** | **0.2776±0.0013** |
| Weather | 96 | 1.4216±0.2777 | 0.8670±0.0771 | **0.6420±0.0319** | **0.5535±0.0118** | 0.1975±0.0013 | 0.2577±0.0030 | **0.1758±0.0023** | **0.2435±0.0038** |
| | 192 | 1.4292±0.3063 | 0.8800±0.0936 | **0.8767±0.0966** | **0.6636±0.0304** | 0.2371±0.0005 | 0.2963±0.0010 | **0.2194±0.0041** | **0.2795±0.0013** |
| | 336 | 1.7964±0.6657 | 1.0076±0.1761 | **1.5063±0.1426** | **0.8511±0.0366** | 0.2819±0.0005 | 0.3331±0.0009 | **0.2652±0.0088** | **0.3121±0.0038** |
| | 720 | 1.5424±0.0315 | 0.9457±0.0208 | **1.4271±0.0529** | **0.8531±0.0099** | 0.3427±0.0004 | 0.3794±0.0002 | **0.3297±0.0010** | **0.3596±0.0006** |

| Method | | TimesNet | | + Ours | | iTransformer | | + Ours | |
|---|---|---|---|---|---|---|---|---|---|
| | | MSE | MAE | MSE | MAE | MSE | MAE | MSE | MAE |
| Electricity | 96 | 0.1753±0.0010 | 0.2799±0.0002 | **0.1541±0.0016** | **0.2475±0.0019** | 0.1525±0.0001 | 0.2460±0.0004 | **0.1197±0.0035** | **0.1979±0.0006** |
| | 192 | 0.1908±0.0043 | 0.2933±0.0038 | **0.1694±0.0017** | **0.2608±0.0018** | 0.1674±0.0003 | 0.2593±0.0007 | **0.1434±0.0009** | **0.2157±0.0011** |
| | 336 | 0.2112±0.0094 | 0.3104±0.0075 | **0.1850±0.0041** | **0.2762±0.0039** | 0.1830±0.0018 | 0.2762±0.0010 | **0.1683±0.0023** | **0.2395±0.0047** |
| | 720 | 0.2353±0.0063 | 0.3257±0.0028 | **0.2260±0.0112** | **0.3027±0.0033** | 0.2199±0.0007 | 0.3097±0.0010 | **0.2169±0.0039** | **0.2786±0.0005** |
| ETTh1 | 96 | 0.4534±0.0095 | 0.4811±0.0054 | **0.4349±0.0055** | **0.4643±0.0027** | 0.4200±0.0039 | 0.4536±0.0038 | **0.4112±0.0017** | **0.4407±0.0014** |
| | 192 | 0.5331±0.0019 | 0.5308±0.0054 | **0.5195±0.0103** | **0.5170±0.0066** | 0.4944±0.0111 | 0.5020±0.0059 | **0.4739±0.0039** | **0.4816±0.0021** |
| | 336 | 0.6209±0.0027 | 0.5796±0.0013 | **0.5961±0.0014** | **0.5598±0.0004** | 0.5382±0.0035 | 0.5276±0.0020 | **0.5336±0.0033** | **0.5188±0.0031** |
| | 720 | 0.8444±0.0717 | 0.6968±0.0313 | **0.7730±0.0163** | **0.6605±0.0047** | 0.7159±0.0157 | 0.6293±0.0069 | **0.7040±0.0034** | **0.6153±0.0001** |
| ETTh2 | 96 | 0.1831±0.0028 | 0.2980±0.0012 | **0.1740±0.0064** | **0.2764±0.0035** | 0.1769±0.0039 | 0.2871±0.0017 | **0.1720±0.0040** | **0.2708±0.0013** |
| | 192 | 0.2177±0.0020 | 0.3294±0.0030 | **0.2040±0.0036** | **0.3060±0.0017** | 0.1992±0.0019 | 0.3109±0.0021 | **0.1957±0.0029** | **0.2954±0.0028** |
| | 336 | 0.2396±0.0026 | 0.3457±0.0008 | **0.2185±0.0033** | **0.3194±0.0003** | 0.2197±0.0060 | 0.3290±0.0038 | **0.2127±0.0008** | **0.3061±0.0023** |
| | 720 | 0.2805±0.0103 | 0.3764±0.0108 | **0.2779±0.0137** | **0.3634±0.0117** | 0.2708±0.0140 | 0.3661±0.0105 | **0.2698±0.0132** | **0.3562±0.0094** |
| ETTm1 | 96 | 0.4493±0.0265 | 0.4477±0.0117 | **0.3805±0.0109** | **0.3983±0.0036** | 0.3832±0.0071 | 0.4145±0.0033 | **0.3489±0.0065** | **0.3856±0.0041** |
| | 192 | 0.4478±0.0077 | 0.4608±0.0035 | **0.4400±0.0090** | **0.4315±0.0028** | 0.4285±0.0089 | 0.4451±0.0065 | **0.4034±0.0080** | **0.4201±0.0055** |
| | 336 | 0.5502±0.0629 | 0.5041±0.0153 | **0.4944±0.0546** | **0.4616±0.0226** | 0.4851±0.0042 | 0.4794±0.0020 | **0.4680±0.0064** | **0.4604±0.0033** |
| | 720 | 0.6185±0.0334 | 0.5588±0.0167 | **0.5630±0.0406** | **0.5072±0.0139** | 0.5660±0.0049 | 0.5323±0.0024 | **0.5636±0.0055** | **0.5190±0.0041** |
| ETTm2 | 96 | 0.1212±0.0022 | 0.2336±0.0033 | **0.1103±0.0049** | **0.2120±0.0035** | 0.1199±0.0030 | 0.2347±0.0016 | **0.1112±0.0027** | **0.2205±0.0022** |
| | 192 | 0.1550±0.0076 | 0.2665±0.0065 | **0.1358±0.0011** | **0.2385±0.0010** | 0.1491±0.0031 | 0.2656±0.0038 | **0.1438±0.0098** | **0.2517±0.0080** |
| | 336 | 0.1904±0.0070 | 0.2929±0.0038 | **0.1750±0.0041** | **0.2686±0.0016** | 0.1845±0.0026 | 0.2933±0.0025 | **0.1816±0.0017** | **0.2829±0.0009** |
| | 720 | 0.2417±0.0047 | 0.3335±0.0016 | **0.2254±0.0020** | **0.3093±0.0004** | 0.2331±0.0051 | 0.3334±0.0032 | **0.2319±0.0084** | **0.3267±0.0062** |
| Exchange | 96 | 0.0757±0.0030 | 0.1983±0.0035 | **0.0656±0.0041** | **0.1769±0.0044** | 0.0580±0.0025 | 0.1719±0.0037 | **0.0514±0.0028** | **0.1580±0.0049** |
| | 192 | 0.1347±0.0074 | 0.2715±0.0078 | **0.1145±0.0044** | **0.2415±0.0045** | 0.1126±0.0019 | 0.2447±0.0023 | **0.1078±0.0062** | **0.2306±0.0061** |
| | 336 | 0.2367±0.0078 | 0.3631±0.0061 | **0.2187±0.0046** | **0.3364±0.0062** | 0.2095±0.0053 | 0.3391±0.0041 | **0.1963±0.0028** | **0.3144±0.0025** |
| | 720 | 0.6357±0.0185 | 0.6179±0.0094 | **0.5945±0.0266** | **0.5822±0.0168** | 0.5172±0.0134 | 0.5505±0.0076 | **0.5102±0.0111** | **0.5291±0.0050** |
| ILI | 24 | 1.4781±0.2225 | 0.7128±0.0491 | **1.3330±0.0889** | **0.6844±0.0190** | 1.1477±0.0827 | 0.6593±0.0222 | **1.1289±0.0025** | **0.6576±0.0067** |
| | 36 | 1.2944±0.0892 | 0.7476±0.0257 | **1.2039±0.0124** | **0.6948±0.0098** | 1.0608±0.0062 | 0.6951±0.0158 | **1.0391±0.0546** | **0.6822±0.0174** |
| | 48 | 1.2802±0.0381 | 0.7363±0.0064 | **1.2777±0.0329** | **0.7207±0.0179** | 1.2094±0.0227 | 0.7351±0.0063 | **1.1637±0.0268** | **0.7148±0.0094** |
| | 60 | 1.2909±0.0688 | 0.7730±0.0261 | **1.1906±0.0354** | **0.7126±0.0093** | 1.2223±0.1032 | 0.7575±0.0409 | **1.1956±0.0122** | **0.7308±0.0017** |
| Traffic | 96 | 0.3601±0.0085 | 0.3141±0.0055 | **0.3215±0.0060** | **0.2784±0.0056** | 0.3084±0.0014 | 0.2717±0.0018 | **0.2828±0.0017** | **0.2485±0.0004** |
| | 192 | 0.3639±0.0026 | 0.3136±0.0033 | **0.3251±0.0080** | **0.2768±0.0045** | 0.3267±0.0020 | 0.2794±0.0023 | **0.2909±0.0144** | **0.2528±0.0076** |
| | 336 | 0.3734±0.0044 | 0.3200±0.0047 | **0.3308±0.0090** | **0.2824±0.0067** | 0.3381±0.0006 | 0.2850±0.0007 | **0.3005±0.0062** | **0.2594±0.0043** |
| | 720 | 0.3958±0.0056 | 0.3368±0.0054 | **0.3388±0.0049** | **0.2885±0.0050** | 0.3574±0.0010 | 0.3015±0.0010 | **0.3201±0.0064** | **0.2730±0.0049** |
| Weather | 96 | 0.1880±0.0170 | 0.2380±0.0146 | **0.1706±0.0034** | **0.2257±0.0026** | 0.1784±0.0005 | 0.2229±0.0029 | **0.1587±0.0061** | **0.2199±0.0067** |
| | 192 | 0.2344±0.0102 | 0.2775±0.0070 | **0.2331±0.0062** | **0.2772±0.0039** | 0.2308±0.0011 | 0.2675±0.0021 | **0.2138±0.0033** | **0.2654±0.0037** |
| | 336 | 0.2928±0.0048 | 0.3171±0.0036 | **0.2875±0.0062** | **0.3144±0.0053** | 0.2892±0.0019 | 0.3099±0.0030 | **0.2733±0.0019** | **0.3058±0.0004** |
| | 720 | 0.3684±0.0029 | 0.3654±0.0016 | **0.3640±0.0022** | **0.3622±0.0019** | 0.3701±0.0015 | 0.3634±0.0014 | **0.3520±0.0044** | **0.3547±0.0026** |

Table 6: The complete forecasting errors for multivariate time series of ablation study among GLAFF and variants. A lower outcome indicates a better prediction. The best results are highlighted in bold.

| Method | | iTransformer | | + Ours | | w/o Backbone | | w/o Attention | | w/o Quantile | | w/o Adaptive | |
|---|---|---|---|---|---|---|---|---|---|---|---|---|---|
| | | MSE | MAE | MSE | MAE | MSE | MAE | MSE | MAE | MSE | MAE | MSE | MAE |
| Electricity | 96 | 0.1525 | 0.2460 | **0.1197** | **0.1979** | 0.2058 | 0.2663 | 0.1518 | 0.2450 | 0.1467 | 0.2465 | 0.1574 | 0.2502 |
| | 192 | 0.1674 | 0.2593 | **0.1434** | **0.2157** | 0.2097 | 0.2793 | 0.1684 | 0.2610 | 0.1677 | 0.2662 | 0.1740 | 0.2662 |
| | 336 | 0.1830 | 0.2762 | **0.1683** | **0.2395** | 0.2454 | 0.3014 | 0.1832 | 0.2775 | 0.1993 | 0.2954 | 0.1953 | 0.2877 |
| | 720 | 0.2199 | 0.3097 | **0.2169** | **0.2786** | 0.2984 | 0.3386 | 0.2182 | 0.3092 | 0.2593 | 0.3403 | 0.2330 | 0.3171 |
| ETTh1 | 96 | 0.4200 | 0.4536 | **0.4112** | **0.4407** | 0.5741 | 0.5150 | 0.4282 | 0.4560 | 0.4230 | 0.4598 | 0.4590 | 0.4754 |
| | 192 | 0.4944 | 0.5020 | **0.4739** | **0.4816** | 0.6422 | 0.5409 | 0.5166 | 0.5112 | 0.4861 | 0.5019 | 0.5427 | 0.5258 |
| | 336 | 0.5382 | 0.5276 | **0.5336** | **0.5188** | 0.7020 | 0.5833 | 0.5797 | 0.5513 | 0.5496 | 0.5386 | 0.5836 | 0.5469 |
| | 720 | 0.7159 | 0.6293 | **0.7040** | **0.6153** | 0.8481 | 0.6541 | 0.7564 | 0.6442 | 0.7065 | 0.6294 | 0.7429 | 0.6429 |
| ETTh2 | 96 | 0.1769 | 0.2871 | **0.1720** | **0.2708** | 0.2095 | 0.2786 | 0.1941 | 0.3270 | 0.1969 | 0.3308 | 0.2028 | 0.3337 |
| | 192 | 0.1992 | 0.3109 | **0.1957** | **0.2954** | 0.2847 | 0.3274 | 0.2325 | 0.3618 | 0.2223 | 0.3562 | 0.2335 | 0.3631 |
| | 336 | 0.2197 | 0.3290 | **0.2127** | **0.3061** | 0.2937 | 0.3384 | 0.2432 | 0.3722 | 0.2639 | 0.3896 | 0.2611 | 0.3848 |
| | 720 | 0.2708 | 0.3661 | **0.2698** | **0.3562** | 0.3082 | 0.3567 | 0.2909 | 0.4051 | 0.2786 | 0.4036 | 0.3041 | 0.4197 |
| ETTm1 | 96 | 0.3832 | 0.4145 | **0.3489** | **0.3856** | 0.4161 | 0.4100 | 0.4106 | 0.4233 | 0.3595 | 0.4021 | 0.3841 | 0.4180 |
| | 192 | 0.4285 | 0.4451 | **0.4034** | **0.4201** | 0.4932 | 0.4587 | 0.4547 | 0.4522 | 0.4138 | 0.4385 | 0.4397 | 0.4511 |
| | 336 | 0.4851 | 0.4794 | **0.4680** | **0.4604** | 0.5467 | 0.4883 | 0.5278 | 0.4990 | 0.4797 | 0.4806 | 0.4984 | 0.4881 |
| | 720 | 0.5660 | 0.5323 | **0.5636** | **0.5190** | 0.6210 | 0.5316 | 0.5942 | 0.5441 | 0.5814 | 0.5383 | 0.5883 | 0.5400 |
| ETTm2 | 96 | 0.1199 | 0.2347 | **0.1112** | **0.2205** | 0.1334 | 0.2340 | 0.1197 | 0.2337 | 0.1129 | 0.2293 | 0.1150 | 0.2296 |
| | 192 | 0.1491 | 0.2656 | **0.1438** | **0.2517** | 0.1612 | 0.2624 | 0.1554 | 0.2677 | 0.1459 | 0.2627 | 0.1470 | 0.2614 |
| | 336 | 0.1845 | 0.2933 | **0.1816** | **0.2829** | 0.1979 | 0.2901 | 0.1884 | 0.2944 | 0.1935 | 0.3006 | 0.1837 | 0.2924 |
| | 720 | 0.2331 | 0.3334 | **0.2319** | **0.3267** | 0.2464 | 0.3272 | 0.2332 | 0.3297 | 0.2791 | 0.3627 | 0.2324 | 0.3306 |
| Exchange | 96 | 0.0580 | 0.1719 | **0.0514** | **0.1580** | 0.0972 | 0.2212 | 0.0595 | 0.1748 | 0.0613 | 0.1771 | 0.0669 | 0.1870 |
| | 192 | 0.1126 | 0.2447 | **0.1078** | **0.2306** | 0.1448 | 0.2779 | 0.1149 | 0.2475 | 0.1154 | 0.2470 | 0.1203 | 0.2539 |
| | 336 | 0.2095 | 0.3391 | **0.1963** | **0.3144** | 0.2614 | 0.3772 | 0.2079 | 0.3371 | 0.2081 | 0.3359 | 0.2129 | 0.3404 |
| | 720 | 0.5172 | 0.5505 | **0.5102** | **0.5291** | 0.6089 | 0.5952 | 0.5308 | 0.5582 | 0.5359 | 0.5605 | 0.5461 | 0.5668 |
| ILI | 24 | 1.1477 | 0.6593 | **1.1289** | **0.6576** | 2.3361 | 1.0211 | 1.1321 | 0.6759 | 1.2641 | 0.6907 | 1.4264 | 0.7593 |
| | 36 | 1.0608 | 0.6951 | **1.0391** | **0.6822** | 1.9812 | 0.9408 | 1.1258 | 0.7193 | 1.0839 | 0.7035 | 1.2890 | 0.7669 |
| | 48 | 1.2094 | 0.7351 | **1.1637** | **0.7148** | 1.8142 | 0.9166 | 1.2250 | 0.7509 | 1.2472 | 0.7483 | 1.3530 | 0.7789 |
| | 60 | 1.2223 | 0.7575 | **1.1956** | **0.7308** | 1.5257 | 0.8484 | 1.2615 | 0.7734 | 1.2239 | 0.7551 | 1.2319 | 0.7593 |
| Traffic | 96 | 0.3084 | 0.2717 | **0.2828** | **0.2485** | 0.3348 | 0.2723 | 0.3172 | 0.2806 | 0.2909 | 0.2684 | 0.2930 | 0.2612 |
| | 192 | 0.3267 | 0.2794 | **0.2909** | **0.2528** | 0.3387 | 0.2736 | 0.3357 | 0.2884 | 0.2948 | 0.2737 | 0.2970 | 0.2610 |
| | 336 | 0.3381 | 0.2850 | **0.3005** | **0.2594** | 0.3460 | 0.2794 | 0.3482 | 0.2958 | 0.3023 | 0.2804 | 0.3082 | 0.2706 |
| | 720 | 0.3574 | 0.3015 | **0.3201** | **0.2730** | 0.3558 | 0.2906 | 0.3684 | 0.3113 | 0.3249 | 0.2984 | 0.3212 | 0.2819 |
| Weather | 96 | 0.1784 | 0.2229 | **0.1587** | **0.2199** | 0.2382 | 0.2695 | 0.1780 | 0.2214 | 0.1811 | 0.2270 | 0.1914 | 0.2379 |
| | 192 | 0.2308 | 0.2675 | **0.2138** | **0.2654** | 0.2882 | 0.3105 | 0.2383 | 0.2733 | 0.2364 | 0.2768 | 0.2489 | 0.2832 |
| | 336 | 0.2892 | 0.3099 | **0.2733** | **0.3058** | 0.3381 | 0.3414 | 0.2932 | 0.3146 | 0.2905 | 0.3134 | 0.3070 | 0.3251 |
| | 720 | 0.3701 | 0.3634 | **0.3520** | **0.3547** | 0.4011 | 0.3813 | 0.3752 | 0.3664 | 0.3727 | 0.3649 | 0.3829 | 0.3722 |
| Avg. | | 0.3957 | 0.3875 | **0.3794** | **0.3689** | 0.5291 | 0.4278 | 0.4100 | 0.3987 | 0.4027 | 0.3971 | 0.4243 | 0.4036 |

## B.3 Hyperparameter Analysis

To assess the sensitivity of GLAFF for various hyperparameter configurations, we conduct a comprehensive hyperparameter analysis. It is worth noting that, in theory, GLAFF introduces only one additional core hyperparameter, the quantile $q$ in the Robust Denormalizer. Therefore, conducting hyperparameter selection for GLAFF requires only very little work. However, to comprehensively compare the effects for different parameter configurations, we also explore the number $l$ of attention blocks and the proportion $p$ of dropout in the Attention-based Mapper. We implement our method on the iTransformer backbone. The length of the history window is fixed at 96, while the length of the prediction window is set to 192. The experimental results for the eight data-rich datasets are depicted in Figure 4.

The quantile in the Robust Denormalizer impacts the ability of GLAFF to adapt to data drift, particularly when anomalies are present within the sliding window. Higher quantiles can render GLAFF overly sensitive to magnitude, diminishing its robustness against outlier data. Conversely, smaller quantiles may hinder the ability of GLAFF to detect distribution changes, consequently impairing its capacity to adapt to data drift. For datasets ETTh1, ETTm1, and Traffic, higher quantiles correlate with increased prediction accuracy, while for datasets ETTh2 and Weather, lower quantiles yield higher prediction accuracy. Moreover, datasets ETTm2, Electricity, and Exchange demonstrate relative robustness to quantile variations. To equalize prediction accuracy, we adopt a quantile $q$ of 0.75 for all datasets and backbones.

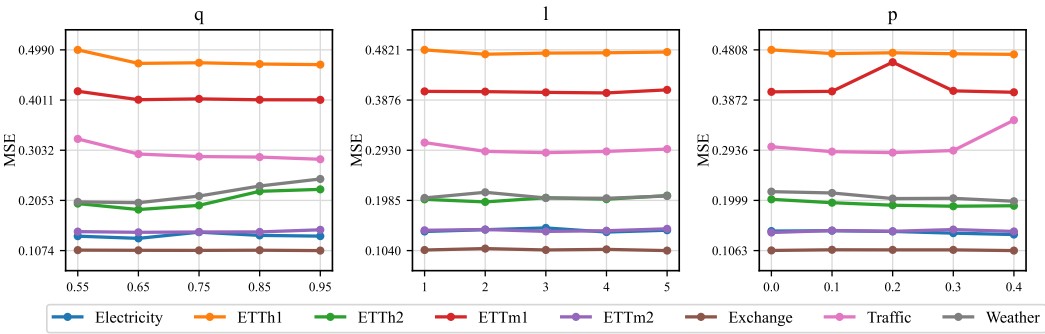

Figure 4: The forecasting errors for multivariate time series of hyperparameter analysis among different configurations for GLAFF. A lower outcome indicates a better prediction.

The number of attention blocks in the Attention-based Mapper influences the modeling process for global information representation. A greater depth in attention blocks enables GLAFF to model timestamp dependencies more sufficiently and uncover latent high-level semantic features. Nevertheless, augmenting the depth of the network also escalates computation costs and exacerbates convergence challenges. According to experimental findings, all benchmark datasets exhibit robustness to variations in the number $l$ of attention blocks, thus facilitating the deployment of GLAFF. Considering training cost and model performance, we choose 2 as the number $l$ of attention blocks for all datasets and backbones.

The proportion of dropout in the Attention-based Mapper impacts the generalization performance of GLAFF. When the dropout proportion is too small, indicating the retention of too many neurons, the model tends to overfit, thereby diminishing its generalization capacity. Conversely, if the dropout proportion is excessively large, meaning the exclusion of too many neurons, the model may struggle to effectively capture the underlying features, decreasing prediction accuracy. Moreover, a high dropout proportion can also impede the convergence of model training due to the varying network structures observed in each training sample. From the experimental results, except for the ETTm1 and Traffic datasets, most datasets exhibit insensitivity to the choice of dropout proportion $p$. To simplify the parameter selection challenge, we adopt a dropout proportion $p$ of 0.1 for all datasets and backbones.

### B.4 Efficiency Evaluation

To assess the practical applicability of GLAFF in real-world environments, we comprehensively compare training time and memory usage between baseline models and GLAFF across nine datasets. The experimental results are summarized in Table 7. Specifically, incorporating GLAFF results in an average 23.4382s increase in the training time and a 25.0608MB increase in memory usage. Presently, with hardware resources evolving rapidly, this computation costs may not affect the training and deployment of GLAFF in most scenarios, particularly when considering the significant accuracy enhancement it offers. Moreover, it is a common practice to trade off computation costs for enhanced prediction accuracy. For instance, the TimesNet baseline exhibits an average training time increase of 355s and an average memory cost increase of 145MB compared to the DLinear baseline, yielding a mere 8.4% increase in average prediction accuracy. In contrast, our GLAFF incurs a training time increase of 14s and a memory usage increase of 25MB while achieving a 13.1% increase in prediction accuracy compared to the DLinear backbone. In scenarios demanding high prediction accuracy, this computation costs are acceptable.

Notably, owing to the unique forecasting mechanism (mapping), the memory usage incurred by GLAFF remains insensitive to both the dataset and the prediction length, consistently hovering at approximately 25MB. Additionally, we observe that the introduction of GLAFF not only results in substantial enhancements in prediction accuracy in all scenarios, but also significantly reduces the training time in some scenarios (such as the TimesNet backbone on the ETTh2 dataset, the Informer backbone on the Traffic dataset, the DLinear backbone, and the TimesNet backbone). We postulate

Table 7: The training time and memory usage for multivariate time series forecasting among GLAFF and mainstream baselines. A lower outcome indicates a better efficiency.

| Method | | Informer Time (s) | Size (MB) | + Ours Time (s) | Size (MB) | DLinear Time (s) | Size (MB) | + Ours Time (s) | Size (MB) | TimesNet Time (s) | Size (MB) | + Ours Time (s) | Size (MB) | iTransformer Time (s) | Size (MB) | + Ours Time (s) | Size (MB) |
|---|---|---|---|---|---|---|---|---|---|---|---|---|---|---|---|---|---|
| Electricity | 96 | 15.92 | 67.05 | 17.63 | 92.52 | 5.701 | 0.071 | 8.226 | 25.54 | 86.18 | 146.2 | 100.5 | 171.6 | 27.71 | 48.48 | 31.49 | 73.95 |
| | 192 | 20.84 | 67.05 | 22.50 | 92.52 | 7.663 | 0.142 | 11.74 | 25.61 | 161.9 | 146.2 | 162.1 | 171.7 | 29.37 | 48.67 | 34.86 | 74.14 |
| | 336 | 29.05 | 67.05 | 31.40 | 92.52 | 10.58 | 0.249 | 17.49 | 25.72 | 396.2 | 146.3 | 279.5 | 171.7 | 31.82 | 48.95 | 40.43 | 74.42 |
| | 720 | 50.30 | 67.05 | 60.02 | 92.52 | 18.88 | 0.533 | 35.79 | 26.00 | 489.2 | 146.4 | 433.4 | 171.9 | 38.24 | 49.70 | 58.18 | 75.17 |
| ETTh1 | 96 | 5.400 | 62.76 | 8.828 | 87.61 | 0.394 | 0.071 | 3.708 | 24.93 | 82.26 | 145.5 | 89.64 | 170.4 | 2.057 | 48.48 | 5.486 | 73.34 |
| | 192 | 6.195 | 62.76 | 11.55 | 87.61 | 0.447 | 0.142 | 5.528 | 25.00 | 137.5 | 145.6 | 169.0 | 170.4 | 2.107 | 48.67 | 7.191 | 73.53 |
| | 336 | 8.104 | 62.76 | 16.49 | 87.61 | 0.543 | 0.249 | 8.687 | 25.10 | 274.1 | 145.6 | 312.7 | 170.5 | 2.095 | 48.95 | 10.25 | 73.81 |
| | 720 | 12.78 | 62.76 | 30.18 | 87.61 | 0.727 | 0.533 | 18.44 | 25.39 | 336.6 | 145.8 | 314.0 | 170.6 | 2.269 | 49.70 | 19.81 | 74.56 |
| ETTh2 | 96 | 5.605 | 62.76 | 8.833 | 87.61 | 0.405 | 0.071 | 3.740 | 24.93 | 98.52 | 145.5 | 105.6 | 170.4 | 1.945 | 48.48 | 5.476 | 73.34 |
| | 192 | 6.155 | 62.76 | 11.60 | 87.61 | 0.459 | 0.142 | 5.517 | 25.00 | 179.3 | 145.6 | 171.1 | 170.4 | 2.027 | 48.67 | 7.214 | 73.53 |
| | 336 | 8.125 | 62.76 | 16.51 | 87.61 | 0.535 | 0.249 | 8.702 | 25.10 | 375.4 | 145.6 | 221.7 | 170.5 | 2.084 | 48.95 | 10.30 | 73.81 |
| | 720 | 12.73 | 62.76 | 30.10 | 87.61 | 0.710 | 0.533 | 18.45 | 25.39 | 372.7 | 145.8 | 299.7 | 170.6 | 2.072 | 49.70 | 19.81 | 74.56 |
| ETTm1 | 96 | 21.43 | 62.76 | 35.98 | 87.61 | 1.550 | 0.071 | 15.14 | 24.93 | 293.0 | 145.5 | 286.9 | 170.4 | 7.929 | 48.48 | 21.96 | 73.34 |
| | 192 | 25.07 | 62.76 | 47.45 | 87.61 | 1.833 | 0.142 | 22.47 | 25.00 | 382.0 | 145.6 | 556.5 | 170.4 | 8.737 | 48.67 | 29.40 | 73.53 |
| | 336 | 33.71 | 62.76 | 68.27 | 87.61 | 2.096 | 0.249 | 35.95 | 25.10 | 1017 | 145.6 | 1296 | 170.5 | 8.003 | 48.95 | 42.53 | 73.81 |
| | 720 | 54.73 | 62.76 | 129.8 | 87.61 | 3.099 | 0.533 | 79.09 | 25.39 | 1530 | 145.8 | 1405 | 170.6 | 8.719 | 49.70 | 84.83 | 74.56 |
| ETTm2 | 96 | 21.83 | 62.76 | 35.44 | 87.61 | 1.553 | 0.071 | 15.11 | 24.93 | 306.7 | 145.5 | 308.4 | 170.4 | 8.144 | 48.48 | 22.09 | 73.34 |
| | 192 | 25.12 | 62.76 | 47.47 | 87.61 | 1.801 | 0.142 | 22.58 | 25.00 | 393.7 | 145.6 | 786.6 | 170.4 | 8.590 | 48.67 | 29.54 | 73.53 |
| | 336 | 33.41 | 62.76 | 68.43 | 87.61 | 2.126 | 0.249 | 35.93 | 25.10 | 1429 | 145.6 | 2631 | 170.5 | 8.318 | 48.95 | 42.44 | 73.81 |
| | 720 | 55.16 | 62.76 | 129.6 | 87.61 | 3.177 | 0.533 | 79.13 | 25.39 | 1499 | 145.8 | 1470 | 170.6 | 9.696 | 49.70 | 84.85 | 74.56 |
| Exchange | 96 | 2.242 | 62.77 | 3.823 | 87.63 | 0.173 | 0.071 | 1.600 | 24.93 | 32.51 | 145.5 | 37.20 | 170.4 | 0.865 | 48.48 | 2.336 | 73.34 |
| | 192 | 2.580 | 62.77 | 4.929 | 87.63 | 0.188 | 0.142 | 2.301 | 25.00 | 66.52 | 145.6 | 66.49 | 170.4 | 0.928 | 48.67 | 3.031 | 73.53 |
| | 336 | 3.340 | 62.77 | 6.752 | 87.63 | 0.218 | 0.249 | 3.543 | 25.11 | 159.0 | 145.6 | 209.2 | 170.5 | 0.859 | 48.95 | 4.209 | 73.81 |
| | 720 | 4.852 | 62.77 | 11.46 | 87.63 | 0.280 | 0.533 | 7.005 | 25.39 | 144.4 | 145.8 | 138.3 | 170.6 | 0.797 | 49.70 | 7.563 | 74.56 |
| ILI | 24 | 0.245 | 62.76 | 0.343 | 87.14 | 0.019 | 0.007 | 0.128 | 24.39 | 1.608 | 145.5 | 1.650 | 169.9 | 0.162 | 48.22 | 0.253 | 72.61 |
| | 36 | 0.244 | 62.76 | 0.326 | 87.14 | 0.019 | 0.010 | 0.129 | 24.40 | 1.664 | 145.5 | 1.717 | 169.9 | 0.146 | 48.25 | 0.252 | 72.63 |
| | 48 | 0.246 | 62.76 | 0.336 | 87.14 | 0.018 | 0.014 | 0.132 | 24.40 | 1.688 | 145.5 | 1.712 | 169.9 | 0.146 | 48.27 | 0.249 | 72.66 |
| | 60 | 0.248 | 62.76 | 0.354 | 87.14 | 0.019 | 0.017 | 0.143 | 24.40 | 1.769 | 145.5 | 1.701 | 169.9 | 0.141 | 48.30 | 0.270 | 72.68 |
| Traffic | 96 | 18.17 | 74.45 | 14.97 | 101.0 | 8.166 | 0.071 | 8.067 | 26.60 | 75.01 | 147.2 | 74.67 | 173.7 | 60.80 | 48.48 | 62.07 | 75.01 |
| | 192 | 25.29 | 74.45 | 19.77 | 101.0 | 12.17 | 0.142 | 11.49 | 26.67 | 131.2 | 147.3 | 102.1 | 173.8 | 63.71 | 48.67 | 64.85 | 75.20 |
| | 336 | 37.10 | 74.45 | 27.89 | 101.0 | 17.77 | 0.249 | 16.94 | 26.78 | 189.1 | 147.3 | 177.3 | 173.8 | 67.71 | 48.95 | 69.42 | 75.48 |
| | 720 | 65.19 | 74.45 | 49.28 | 101.0 | 31.00 | 0.533 | 32.11 | 27.06 | 266.4 | 147.5 | 286.0 | 174.0 | 77.75 | 49.70 | 82.50 | 76.23 |
| Weather | 96 | 16.69 | 62.95 | 27.68 | 87.83 | 1.645 | 0.071 | 11.71 | 24.95 | 208.8 | 145.6 | 274.0 | 170.5 | 6.877 | 48.48 | 17.06 | 73.37 |
| | 192 | 19.88 | 62.95 | 35.94 | 87.83 | 1.988 | 0.142 | 17.23 | 25.02 | 363.6 | 145.6 | 500.2 | 170.5 | 7.318 | 48.67 | 23.05 | 73.55 |
| | 336 | 26.84 | 62.95 | 51.92 | 87.83 | 2.472 | 0.249 | 27.46 | 25.13 | 653.6 | 145.7 | 743.1 | 170.5 | 7.614 | 48.95 | 32.99 | 73.84 |
| | 720 | 43.84 | 62.95 | 98.33 | 87.83 | 3.918 | 0.533 | 59.82 | 25.42 | 785.8 | 145.8 | 786.7 | 170.7 | 8.277 | 49.70 | 64.74 | 74.59 |

that this phenomenon can be attributed to the incorporation of global information, which accelerates the convergence of the network.

### B.5 Prediction Showcase

In addition to evaluation metrics, forecasting quality is crucial. To comprehensively compare GLAFF and the four mainstream forecasting models, we present the complete prediction showcases for the nine real-world datasets in Figure 5. For the ILI dataset, the length of the history window is set to 36, while the length of the prediction window is set to 48. Except for the ILI dataset, the length of the history window is set to 96, and the length of the prediction window is set to 192. We can observe that by fusing the robustness of the global information and the flexibility of the local information, GLAFF demonstrates superior suitability for intricate and fluctuating real-world scenarios, thereby enhancing the ability of the backbone model to generate predictions that closely correspond to the ground truth.

## C Broader Impact

The GLAFF proposed in our paper focuses on leveraging global information, as denoted by timestamps, to enhance the robust prediction capability of time series forecasting models in the real world. As a model-agnostic and plug-and-play module, GLAFF significantly improves mainstream forecasting models, positively influencing domains such as finance, transportation, energy, healthcare, climate, etc. Furthermore, GLAFF may inspire the community to give more attention to the utilization of global information and catalyze further development of time series forecasting techniques. The source code and checkpoints have been made publicly available to support future research. This paper only focuses on the algorithm design. Using all the codes and datasets strictly follows the

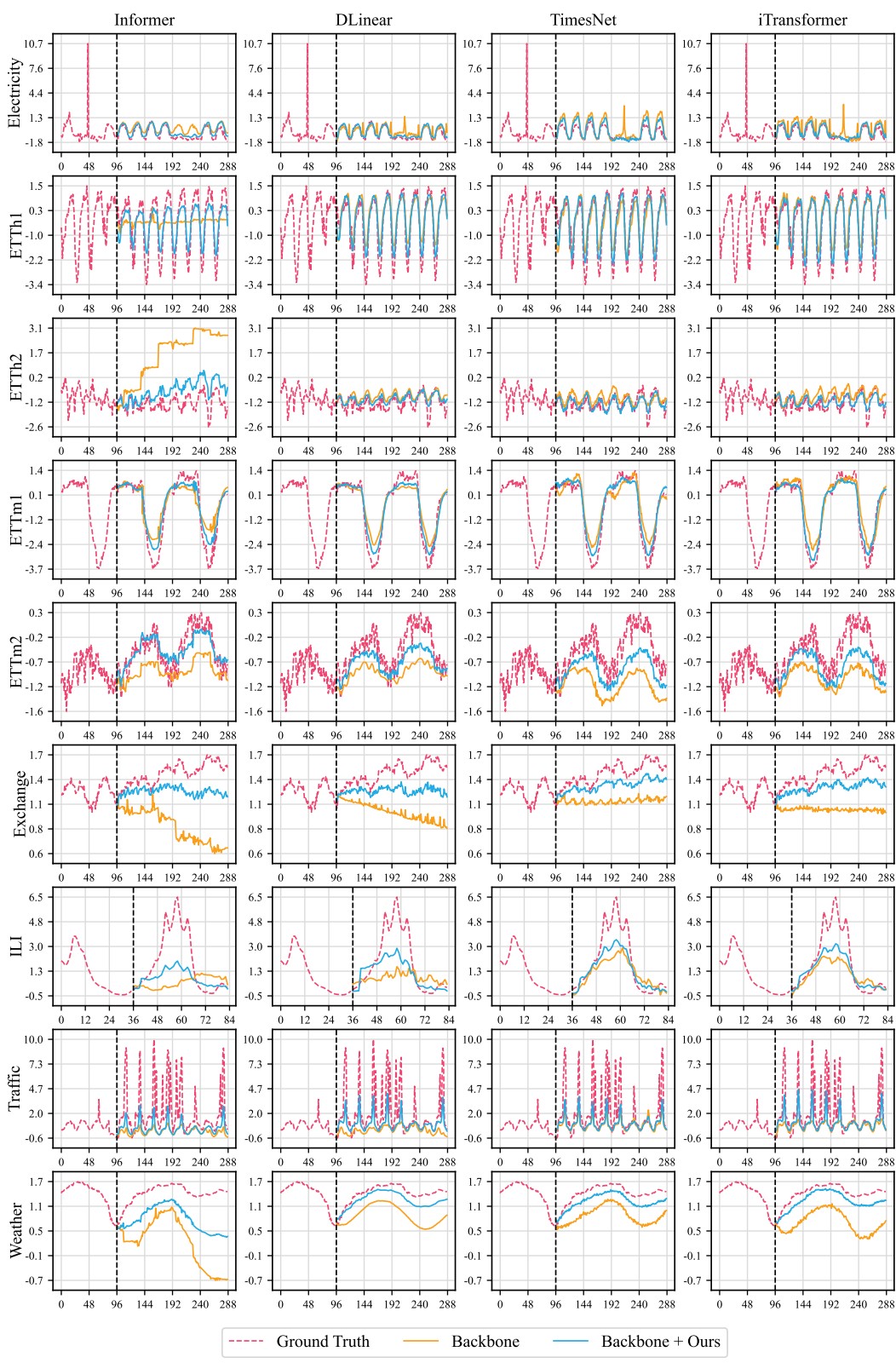

Figure 5: The complete illustration of prediction showcases among GLAFF and mainstream baselines.

corresponding licenses (Appendix A.1 and Appendix A.2). There is no potential ethical risk or negative social impact.

## D   Limitation

While GLAFF exhibits encouraging performance on benchmark datasets, it is subject to certain limitations. As a model-agnostic and plug-and-play framework, GLAFF incurs considerable computation costs attributed to the utilization of stacked attention blocks in Attention-based Mapper (Appendix B.4). Presently, with hardware resources evolving rapidly, this computation costs may not affect the training and deployment of GLAFF in most scenarios. However, GLAFF will likely encounter operational challenges in resource-constrained edge devices, thus restricting its applicability. In future work, we plan to explore lighter weight and more efficient architectures, such as dilated convolutional or graph neural networks, to replace the conventional attention mechanism.

