# OpenReview forum: "Rethinking the Power of Timestamps for Robust Time Series Forecasting: A Global-Local Fusion Perspective"
_NeurIPS.cc/2024/Conference — NeurIPS 2024 poster_

### Official Review · Reviewer_N7JP · 2024-06-30

**Soundness:** 3
**Presentation:** 3
**Contribution:** 3
**Rating:** 7
**Confidence:** 5

**Summary:**

This article focuses on enhancing time series forecasting capabilities using timestamps and introduces a plug-and-play module called GLAFF. Overall, GLAFF is designed to be simple and lightweight, significantly improving the predictive performance of existing time series forecasting algorithms such as ITransformer and DLinear.

**Strengths:**

1. The article is well-structured, and the writing is clear and appropriate.
2. The motivation is explicit, and the method is straightforward and efficient. The core code is provided in the appendix for readers' convenience.
3. The experiments are relatively comprehensive, involving both large language models and task-specific general models.

**Weaknesses:**

1. The article contains some typos, such as "damaged" in line 32 and "aimed" in line 52.
2. Formula 3 uses quantile for denormalization, but the reason for this choice is not explained. Why is quantile better than std for this purpose?
3. Observing Figure 3, I notice that while GLAFF improves the forecasting performance, it still fails to fully capture the traffic spikes. Can you explain why this is the case and suggest any other potential solutions to address this issue?
4. For the ablation experiment w/o Quantile, I would like to see the results of completely removing robust denormalization.

**Questions:**

Please refer to the weaknesses section.

**Limitations:**

The authors have discussed the limitations of the article in the appendix.

---

> ### Author Rebuttal · Authors · 2024-08-05
>
> Thanks for your positive comments and insightful suggestions. Please find our response below.
>
> **Q1: Some typos in the paper.**
>
> We apologize for the oversight that led to grammatical errors in the paper, causing issues with your reading. These typos will be corrected in the final version.
>
> **Q2: Reasons for using quantile.**
>
> As demonstrated in the Electricity dataset in Figure 3, history windows may contain anomalous noise, such as spikes, due to various complex factors in the real world. The mean and variance are highly sensitive to such spiky noise and tend to misnormalize the output of the mapper to an incorrect distribution. In contrast, quantiles are more robust to such noise and align better with our initial objective of providing robust global information. This is also confirmed by the ablation experiments in Section 4.4.
>
> **Q3: Challenges in capturing traffic spikes.**
>
> We visualize the full traffic flow **in Figure 2 of the Global Response Attachment PDF**. The test set has demonstrated drift, with peaks significantly higher in the latter half compared to the training set. Unless the history window incorporates relevant information (e.g., sharper peaks) or the model undergoes additional adaptive training, resolving this issue is challenging.
>
> **Q4: Additional results for ablation experiments.**
>
> We supplement the ablation experiments based on the iTransformer backbone across two representative datasets. The history window is set to 96, and the prediction window to 192. The results about MSE are presented in the table below. The results underscore the necessity of denormalization and the superiority of employing quartiles for robust denormalization.
>
> |   Dataset   | iTransformer | + Ours | w/o Quantile | w/o Denormalization |
> | :---------: | :----------: | :----: | :----------: | :-----------------: |
> |   Traffic   |    0.3267    | 0.2909 |    0.2948    |       0.3233        |
> | Electricity |    0.1674    | 0.1434 |    0.1677    |       0.1742        |

---

> > ### Comment · Reviewer_N7JP · 2024-08-08
> >
> > The authors have effectively addressed  my concerns. Based on these revisions and your thorough response, I have raised my score to 7. I  recommend accepting this paper,

---

> > > ### Author Response · Authors · 2024-08-09
> > >
> > > Thank you for thoroughly reviewing our rebuttal and deciding to raise the rating score. We appreciate your consideration and the time you have dedicated to evaluating our work.

---

### Official Review · Reviewer_Y7JY · 2024-07-11

**Soundness:** 4
**Presentation:** 3
**Contribution:** 2
**Rating:** 6
**Confidence:** 3

**Summary:**

This paper introduces the GLAFF framework where time series models are adapted to also capture "global" information by using information content in the datetime parsed in components of habitual meaning to complement baseline models ("backbones").  The global information is represented through a Mapper, quantile-based Denormalizer, and Combiner, and is combined in an MLP head with the backbone output to form the prediction.  While extracting the time units from datetimes is used in other NLP literatures, it has not been applied in time series forecasting models as far as I am aware.

**Strengths:**

The work appears to be original and augments leading time series models, leading to performance improvements on a suite of time series prediction benchmarks as compared to the respective baseline models without GLAFF.  The baseline models held previous SOTA performances in the last 5 years or so.  Relevant alternatives (GPT2, Table 2) and relevant ablations are done for the best performing baseline iTransformer (Table 3).  The results are presented clearly, with full tables of results in the appendix.

**Weaknesses:**

The intuition paragraphs describing the motivation for the components could be made clearer, in particular with respect to "mitigating data drift"? What assumptions are made explicitly, and what is the precise formulation?

The GLAFF framework could have be applied to a larger set of time series models, or at least applied to the best performing one for each dataset, since they are readily accessible and compared against: (e.g. https://github.com/thuml/Time-Series-Library).

Hyperparameter searches were done on axes rather than a grid.

**Questions:**

See first two limitations above.The

**Limitations:**

The limitation section is in the appendix and describe computational cost.  So the limitations section should be expanded upon, e.g. with respect to moving away from small benchmark datasets, irregular sampling, data drift settings, etc.

---

> ### Author Rebuttal · Authors · 2024-08-05
>
> Thanks for your positive comments and insightful suggestions. Please find our response below.
>
> **Q1: Explanation of the motivation for the component.**
>
> We apologize for the confusion. We will further explain the role of the denormalizer in mitigating data drift. The statistical characteristics (e.g., mean or variance) of time series data collected from the real world typically change over time as the system evolves dynamically. For instance, an increase in the popularity of a service can cause customer metrics (e.g., request counts) to rise over time. Previous studies [1, 2] have shown that models trained on the training set may underperform on the test set when the distributional difference between the training and test sets is significant. We assume that the difference between the distributions of the history and prediction windows is negligible within a small sliding window. As illustrated in Equation 3, the inverse denormalizer mitigates data drift by aligning the output of the mapper to the distribution of the local history window, thereby smoothing the difference between the training and test sets.
>
> **Q2: Additional results for more backbones.**
>
> The principle for selecting the backbone in our study is to balance model architecture, prediction accuracy, and timestamp treatment within a constrained space. Based on the library you provided, the current top three rankings for long-term forecasting are iTransformer, TimeMixer, and TimesNet. iTransformer and TimesNet have already been discussed in our paper, and we now supplement experimental results for TimeMixer. The history window is set to 96 and the prediction window to 192 for all datasets except ILI, where the history and prediction windows are both set to 36. The MSE are presented in the table below. The results demonstrate that our proposed method enhances performance across various mainstream backbones.
>
> |   Dataset   | iTransformer | + Ours | TimeMixer | + Ours | TimesNet | + Ours |
> | :---------: | :----------: | :----: | :-------: | :----: | :------: | :----: |
> | Electricity |    0.1674    | 0.1434 |  0.1890   | 0.1576 |  0.1908  | 0.1694 |
> |    ETTh1    |    0.4944    | 0.4739 |  0.4978   | 0.4897 |  0.5331  | 0.5195 |
> |    ETTh2    |    0.1992    | 0.1957 |  0.2041   | 0.2028 |  0.2177  | 0.2040 |
> |    ETTm1    |    0.4285    | 0.4034 |  0.4429   | 0.4282 |  0.4478  | 0.4400 |
> |    ETTm2    |    0.1491    | 0.1438 |  0.1443   | 0.1393 |  0.1550  | 0.1358 |
> |  Exchange   |    0.1126    | 0.1078 |  0.1207   | 0.1174 |  0.1347  | 0.1145 |
> |     ILI     |    1.0608    | 1.0391 |  1.2741   | 1.2252 |  1.2944  | 1.2039 |
> |   Traffic   |    0.3267    | 0.2909 |  0.3405   | 0.3009 |  0.3639  | 0.3251 |
> |   Weather   |    0.2308    | 0.2138 |  0.2337   | 0.2246 |  0.2344  | 0.2331 |
> |    Avg.     |    0.3522    | 0.3346 |  0.3830   | 0.3651 |  0.3969  | 0.3717 |
>
> **References**
>
> [1] 2022, Reversible Instance Normalization for Accurate Time-Series Forecasting Against Distribution Shift
>
> [2] 2023, Adaptive Normalization for Non-stationary Time Series Forecasting: a Temporal Slice Perspective

---

> ### Author Response · Authors · 2024-08-12
>
> Dear Reviewer Y7JY,
>
> We greatly appreciate the time and effort you have invested in reviewing our paper and providing insightful feedback. As a gentle reminder, it has been more than 5 days since we submitted our rebuttal. As the discussion period is drawing to a close, we wish to ensure that our rebuttal has comprehensively addressed your concerns. We are keen to receive any further feedback you might have and are prepared to make additional clarifications or modifications as needed. Thank you once again for your valuable insights. We look forward to your final thoughts.

---

> > ### Comment · Reviewer_Y7JY · 2024-08-12
> >
> > The additional experiments further back the method and are appreciated.  I remain with minor concerns about the exposition of the intuition/motivation which I believe could be further tightened, though I believe these could be addressed prior to a camera ready version.

---

> > > ### Author Response · Authors · 2024-08-13
> > >
> > > Thank you for carefully reviewing our rebuttal and actively providing feedback. We will present a more tightened exposition of the motivation for this paper.
> > >
> > > > Time series forecasting is vital across various domains. However, existing models predominantly rely on local observations and inadequately utilize the extensive global information embedded in timestamps. This oversight reduces the robustness of these models, particularly when real-world data is noisy or contains anomalies. To address this issue, we propose GLAFF, an innovative framework that more comprehensively integrates timestamp information through late fusion (decision-level fusion), thereby enhancing the accuracy and robustness of time series forecasting backbones.
> > >
> > > Regarding late fusion, we provide the following explanation.
> > >
> > > > Early fusion (Informer) integrates modalities into a single representation at the input level and processes the fused representation through the model. Late fusion (GLAFF) allows each modality to run independently through its own model and fuses the outputs of each modality. Compared to early fusion, late fusion maximizes the processing effectiveness of each modality and is less susceptible to the noise of a single modality, resulting in greater robustness and reliability.
> > >
> > > We hope that these clarifications have addressed your concerns. Should you have any further concerns or questions, please do not hesitate to contact us.

---

> > > ### Author Response · Authors · 2024-08-13
> > >
> > > Dear Reviewer Y7JY,
> > >
> > > We sincerely appreciate the time and effort you have devoted to reviewing our paper and offering valuable feedback. As the discussion period nears its conclusion, we wish to ensure that our rebuttal has thoroughly addressed your concerns. We are eager to receive any additional feedback you may have and are ready to provide further clarifications or make modifications as necessary. Lastly, we look forward to your final comments regarding the score.

---

### Official Review · Reviewer_XBcx · 2024-07-12

**Soundness:** 2
**Presentation:** 3
**Contribution:** 2
**Rating:** 4
**Confidence:** 4

**Summary:**

The paper introduces GLAFF, a novel framework that enhances time series forecasting by modeling timestamps to capture global dependencies and adaptively balancing global and local information, resulting in a significant improvement of 12.5% over existing methods in experiments across nine real-world datasets.

**Strengths:**

1. The writing is clear and easy to understand.
2. Code is provided.
3. Comprehensive experiments consistently enhance performance.

**Weaknesses:**

My concerns are as follows:
1. This paper focuses on utilizing timestamps but only discusses and compares some general time-series forecasting methods. Please discuss the differences and performance comparison with existing methods that focus on better utilizing timestamps, such as methods across temporal scales [1][2] and representation for each timestamp [3].
   - [1] AutoCTS: Automated correlated time series forecasting
   - [2] METRO: a generic graph neural network framework for multivariate time series forecasting
   - [3] TS2Vec: Towards Universal Representation of Time Series

2. The method appears simple and lacks technical contributions, resembling a straightforward dual-pathway combination; it lacks theoretical backing, particularly in theoretical analysis of existing backbones' capabilities with timestamps, and only provides a case study for discussion.

3. I noticed that the adaptive weight has a minimal effect, and I am curious about how the weights in the Combiner are initialized, how they are tuned, and whether they are sensitive.

4. The usage has limitations, requiring high-precision timestamps. Please analyze the impact of timestamp granularity and noise on the results.

5. By simply increasing the lookback window, existing methods usually have better performance (better leverage periodicity). Please analyze the performance improvements under different lookback window lengths, especially longer lengths, to validate more realistic effectiveness.

**Questions:**

Please check concerns

**Limitations:**

The paper does not discuss limitations. Please elaborate more on when the method may perform poorly.

---

> ### Author Rebuttal · Authors · 2024-08-05
>
> Thanks for your valuable comments. We will answer the questions one by one.
>
> **Q1: Comparison with additional baselines.**
>
> Thank you very much for your supplement. All three articles have significant contributions to our field. However, there seems to be some misunderstanding regarding our work. The timestamps mentioned in our paper denote external assistive information, such as "2024-08-01 12:00:00," along with their embeddings. Upon reviewing their papers and codes, it is evident that none of these three works incorporate this type of information, thereby classifying them in the same category as the baseline DLinear. The term "timestamp" used in their papers is more accurately replaced with "timestep", which indicates a concept of temporal position. Generally, these three works do not utilize assistive information represented by timestamps and do not belong to "existing methods that focus on better utilizing timestamps". Furthermore, the superiority of the baselines adopted in our paper (updated to 2024) compared to these three articles (published in 2021) has been extensively validated by previous work[1, 2]. Due to spatial limitations in the response, we will refrain from repeating it here.
>
> **Q2: Theoretical analysis of the proposed method.**
>
> iTransformer does not utilize the timestamp information of the prediction window, which may contribute to its underutilization of timestamps. Therefore, we focus solely on the comparison between the summation scheme represented by Informer and our proposed method, which can be abstracted as early fusion (feature-level fusion) and late fusion (decision-level fusion). Early fusion (Informer) integrates modalities into a single representation at the input level and processes the fused representation through the model. Late fusion (GLAFF) allows each modality to run independently through its own model and fuses the outputs of each modality. Compared to early fusion, late fusion maximizes the processing effectiveness of each modality and is less susceptible to the noise of a single modality, resulting in greater robustness and reliability. This has been validated by extensive previous work [3, 4]. To mitigate the effect of noise and fully exploit the robustness of global information represented by timestamps, our proposed method adopts late fusion. Furthermore, we supplement the ablation results about timestamp across the nine datasets. The MSE results **in Table 1 of the Global Response attachment PDF** clearly indicate that simple fusion methods, such as summation or concatenation, are ineffective with timestamps.
>
> **Q3: Explanation of the Adaptive Combiner.**
>
> Firstly, according to the ablation results on the nine datasets in Appendix B.2, the effect of the Adaptive Combiner is not "minimal", second only to the complete removal of the prediction backbone. Secondly, the combined weight $\mathbf{W}$ of the global mapping $\hat{\mathbf{Y}}$ and local prediction $\bar{\mathbf{Y}}$ in the Adaptive Combiner is derived from the MLP weight generation network. We do not specifically set the initial model weights of the MLP layer. They are entirely adapted autonomously based on the MSE loss using gradient descent from the randomly initialized values. It is important to note that the combination weights $\mathbf{W}$, which cannot be directly initialized, are not the model weights of the MLP.
>
> **Q4: Effects of timestamp granularity and noise.**
>
> Firstly, it must be noted that the timestamp information we integrate has been extensively utilized but ineffectively by various baselines. We do not introduce any supplementary information or assumptions. In other words, our approach does not impose more stringent requirements on timestamps (e.g., higher precision sampling granularity) compared to the Informer (2021) backbone. Secondly, maintaining the stability of prediction models in noisy environments is another extensive area where numerous dedicated works, such as RobustTSF [5], have provided excellent solutions. Nonetheless, to assess the robustness of our proposed method, we present the MSE results of the iTransformer backbone on two representative datasets. The results in the table below clearly demonstrate that our proposed method is robust against varying proportions of Gaussian noise, owing to components such as Robust Denormalizer.
>
> |   Dataset   | iTransformer | + Ours 0% | + Ours 10% | + Ours 20% | + Ours 30% |
> | :---------: | :----------: | :-------: | :--------: | :--------: | :--------: |
> |   Traffic   |    0.3267    |  0.2909   |   0.2932   |   0.2943   |   0.2957   |
> | Electricity |    0.1674    |  0.1434   |   0.1548   |   0.1551   |   0.1555   |
>
> **Q5: Results across varying lookback window lengths.**
>
> We supplement the MSE results on two representative datasets with fixed prediction windows by varying the history windows. The prediction window is fixed at 192. From the experimental results **in Table 2 of the Global Response attachment PDF**, iTransformer and DLinear, which derive their final predictions from a linear layer, demonstrate a general trend of enhanced prediction accuracy with longer history windows. Furthermore, our proposed methods consistently enhance prediction performance.
>
> **Q6: Limitations of the proposed method.**
>
> We have discussed the limitations of our study in Appendix D. The primary emphasis is on the computational cost imposed by the attention mechanism, and a potential solution has been proposed.
>
> **References**
>
> [1] 2024, Self-Supervised Contrastive Learning for Long-Term Forecasting
>
> [2] 2024, Multi-Patch Prediction: Adapting Language Models for Time Series Representation Learning
>
> [3] 2024, Foundations and Trends in Multimodal Machine Learning: Principles, Challenges, and Open Questions
>
> [4] 2023, A Comparative Analysis of Early and Late Fusion for the Multimodal Two-Class Problem
>
> [5] 2024, RobustTSF: Towards Theory and Design of Ro-Bust Time Series Forecasting With Anomalies

---

> ### Author Response · Authors · 2024-08-12
>
> Dear Reviewer XBcx,
>
> We greatly appreciate the time and effort you have invested in reviewing our paper and providing insightful feedback. As a gentle reminder, it has been more than 5 days since we submitted our rebuttal. As the discussion period is drawing to a close, we wish to ensure that our rebuttal has comprehensively addressed your concerns. We are keen to receive any further feedback you might have and are prepared to make additional clarifications or modifications as needed. Thank you once again for your valuable insights. We look forward to your final thoughts.

---

### Official Review · Reviewer_wbSq · 2024-07-12

**Soundness:** 3
**Presentation:** 3
**Contribution:** 3
**Rating:** 6
**Confidence:** 4

**Summary:**

This paper proposes GLAFF which encodes the time stamps of time series and performs self attention across the encodings of the time dimensions, combining it with the output of a global time series forecaster via a learned weighting scheme. As GLAFF is a set of feature constructions, it’s generally additive in performance against any backbone architecture. The main requirement is that GLAFF seems to require knowledge of when the prediction should be produced for.

More specifically, the time stamps themselves are used to generate de-medianed and de-quantized predictions. This is extra side information, and so it should generally tend to be helpful – some of the tested datasets have clear time information. For example, traffic tends to spike in the morning and afternoon hours, while electricity also has daily and monthly peaks and valleys. This is a natural encoding of seasonality and other similar regular events.

**Strengths:**

Originality:

-	Most other papers in this literature tend to focus primarily on architecture, and the ones that do take into account some sort of feature information seem to not incorporate it that well.

-	Overall, I like the idea of separating the time encoding into what’s essentially its own network.

Quality:
-	I appreciate the knockout studies of the varying parts, which is good experimental design.

-	I also appreciate the knockout study of iTransformer and would like to see more of these types of results.

Clarity:

-	Overall, the paper is pretty well written. It’s pretty much clear what’s going on.

Significance:

-	Improving time series forecasting is obviously a highly important problem, and improving the base model is a general purpose technique that should generally be quite useful.

**Weaknesses:**

Quality:

-	My biggest concern in terms of usefulness (and unfortunately, this is somewhat a critique of the entire vein of literature here) is that the time stamps of the prediction window are trained on. This can be, in some sense, a pretty strong lookahead bias. In most deployment settings, we cannot pre-train on the time stamps that we’re going to use to generate the sequence because we do not know the absolute value for the time stamp during train time.

       o	Thus, the method is limited to explicitly semi-regular time series where the forecast clock is known ahead of time. By non-regular forecasting, we can think of wanting to predict the inventory levels of a product after the next sixty sales, or to predict the price of a foreign currency after some amount of trading volume in it. Instead, in regular forecasting, at the start of the day, we may wish to predict the next day’s electricity demands (one of the benchmark datasets).

Clarity:

-	I personally find $\tilde X$, $\hat X$, $\hat Y$ to be quite confusing in terms of notation. Perhaps $\hat T$, $\tilde S$, $\hat S$ could be used instead throughout to denote the fact that these values come from the encoded time stamps.

       o	In line 197, I would encourage against setting the final prediction as $Y$, but rather as $\hat Y$.

-	I also find the methods section somewhat unclear, as it’s a bit tricky to parse out that the median / quantile stats are dependent on the input time series, X. Perhaps it would be better to not separate X and Y entirely, but rather point out that the time series is really cat(X, Y).

-	From the intro and contributions, it could probably be made clear that other papers _do_ consider time stamp information; however, it doesn’t seem to be helpful in knockout studies. This is more of a writing note than something major.

**Questions:**

-	In Eq 3, where do the median and quantiles of Y come from? Do they come from the global set of sequences? This is I think more for clarity at this point but want to confirm.

-	It seems that most of the improvement in exchange and weather, which are not periodic, is driven by improvement in the worst model (informer). In general, should we expect that there is less improvement on aperiodic data because the time embeddings are less helpful?

-	Often in time series prediction tasks, there is other side information, is this naturally best situated to being encoded solely inside the backbone architecture or should it be modeled alongside the time stamps?

-	Tables 1 and 2: I’d suggest using bar charts to make the presentation more engaging, and moving the tables to the appendix.

-	I find it fairly surprising that informer, timesNet, and iTransformer tend to have minimal dropoff when removing the timestamp pieces. Is this lack of dropoff consistent across datasets?

-	Could the authors include an average improvement across methods for a fixed horizon? It would be interesting to see if DLinear improves the most as a result of adding in the time step information.

---

> ### Author Rebuttal · Authors · 2024-08-05
>
> Thanks for your positive comments and insightful suggestions. Please find our response below.
>
> **Q1: Usefulness of the proposed method in non-regular forecasting.**
>
> The initial point to state is that the timestamp information we integrate has been extensively utilized, though ineffectively, by various existing methods. We introduce no additional information or assumptions. Regarding the non-regular forecasting scenario you describe, we consider it a novel situation distinct from the current task. It appears to utilize transaction counts instead of time intervals to denote timing changes. Due to the differing domains, our proposed approach cannot natively support this scenario. As mainstream methods like iTransformer cannot address time series forecasting with missing values, we cannot expect a framework to solve all forecasting challenges. To my knowledge, neither the datasets nor the baselines examined in our paper are relevant to the new scenario you mentioned. Nevertheless, the idea of information fusion that we have proposed may be beneficial. Furthermore, the non-regular forecasting appears intriguing, and we look forward to further discussion after the review period concludes.
>
> **Q2: Confusion caused by the notations.**
>
> We use $\mathbf{S}$ and $\mathbf{T}$ for timestamps, where the last dimension is 6, and $\mathbf{X}$ and $\mathbf{Y}$ for observations, where the last dimension represents the number of channels in the multivariate time series. We apologize for the confusion and will incorporate your suggestion to enhance the readability of our article. Regarding the output in row 197, a marker on $\mathbf{Y}$ is indeed necessary to differentiate it from the actual labels, and we will address this in the final version.
>
> **Q3: Explanation of parsing statistics and Equation 3.**
>
> The median and quantile in Equation 3 are derived from the historical actual observation $\mathbf{X}$ and the historical initial mapping $\tilde{\mathbf{X}}$ within a local sliding window to accommodate the dynamics of the time series. We apply the same statistics, as indicated in the parentheses in Equation 3, to both the historical mapping $\tilde{\mathbf{X}}$ and future mapping $\tilde{\mathbf{Y}}$. We apologize for any confusion, but we do not advise concatenating $\tilde{\mathbf{X}}$ and $\tilde{\mathbf{Y}}$, as the following Adaptive Combiner will require separate historical mapping $\hat{\mathbf{X}}$ and future mapping $\hat{\mathbf{Y}}$ .
>
> **Q4: Performance of the proposed method on aperiodic data.**
>
> Robust global information represented by timestamps indeed offers less assistance for predicting non-periodic data than periodic domains such as transportation and electricity. In fact, this paper primarily focuses on effectively integrating available assistive information. Accurate prediction of dynamic and non-periodic time series is more challenging and requires the inclusion of more assistive information. The information fusion approach we have proposed is also valuable for integrating other supplementary information.
>
> **Q5: Modeling approaches for other side information.**
>
> We should design the modeling method for each type of assistive information based on its characteristics and practical requirements. According to the findings of our paper, simple fusion techniques such as summation or concatenation often fail to fully utilize the assistive information. However, more nuanced modeling generally entails greater resource consumption. In practical applications, it is essential to carefully balance prediction accuracy with computational cost.
>
> **Q6: Enhancing the engagement level of presentations.**
>
> It is true that bar charts provide a more visually engaging presentation of backbone performance improvements. However, arranging various combinations of backbones, metrics, datasets, and prediction lengths within a limited space remains a considerable challenge. We will persist in seeking more effective presentation methods.
>
> **Q7: Impact of timestamps on baseline methods.**
>
> We supplement the ablation results about timestamp across the nine datasets. The history window is set to 96 and the prediction window to 192 for all datasets except ILI, where the history and prediction windows are both set to 36. The MSE results **in Table 1 of the Global Response attachment PDF** clearly indicate that simple fusion methods, such as summation or concatenation, are ineffective with timestamps.
>
> **Q8: Results across varying lookback window lengths.**
>
> We supplement the MSE results on two representative datasets with fixed prediction windows by varying the history windows. The prediction window is fixed at 192. From the experimental results **in Table 2 of the Global Response attachment PDF**, iTransformer and DLinear, which derive their final predictions from a linear layer, demonstrate a general trend of enhanced prediction accuracy with longer history windows. Furthermore, our proposed methods consistently enhance prediction performance.

---

> > ### Comment · Reviewer_wbSq · 2024-08-11
> >
> > Thanks for answering my questions, my opinion of the paper is somewhat improved. I trust the authors will cleanup the notation somewhat in the camera ready and update some of the plots. Overall, I think your approach (and you demonstrate this) does tend to account for time domain information better than existing methods.
> >
> > > As mainstream methods like iTransformer cannot address time series forecasting with missing values, we cannot expect a framework to solve all forecasting challenges.
> >
> > Indeed, my comment is really a critique of the existing literature and probably shouldn't be held too harshly against your work. However, I think it is a straightforward application from your approach (maybe you'd need to forecast when the next time stamp is), and one that could be extremely helpful in my different domain. The problems I suggested are also somewhat less periodic and likely more challenging than many of the benchmarks (hence increased practical utility).
> >
> > Thanks for providing more lookback windows as well in the updated experiments.

---

> > > ### Author Response · Authors · 2024-08-11
> > >
> > > Thank you for meticulously reviewing our rebuttal. The non-regular forecasting scenario you describe is indeed more challenging, practically significant, and highly engaging. We will explore this further in the future. We greatly appreciate your consideration and time devoted to evaluating our work.

---

> > > ### Author Response · Authors · 2024-08-12
> > >
> > > Dear Reviewer wbSq,
> > >
> > > We sincerely appreciate the time and effort you have devoted to reviewing our paper and offering valuable feedback. As the discussion period nears its conclusion, we wish to ensure that our rebuttal has thoroughly addressed your concerns. We are eager to receive any additional feedback you may have and are ready to provide further clarifications or make modifications as necessary. Lastly, we look forward to your final comments regarding the score.

---

### Official Review · Reviewer_NzBw · 2024-07-12

**Soundness:** 3
**Presentation:** 3
**Contribution:** 3
**Rating:** 7
**Confidence:** 4

**Summary:**

The authors proposed a plugin to utilize global information from timestamps in time series forecasting tasks. The proposed plugin consists of three main components: attention-based timestamp mapper, robust denormalizer and adaptive combiner. The authors found that using the proposed plugin in combination with various backbone models helped achieve an average of 12.5% increase in forecasting prediction performance measured as mean square error and mean absolute error on 9 real-world datasets.

**Strengths:**

•	The paper is well structured and easy to follow.

•	The authors compared the proposed method with different timestamps treatments like summation, concatenation and omission.

•	Well-illustrated prediction showcases provide examples of the usefulness of the proposed plugin.

•	The experimental results are comprehensive and impressive. Extensive results on 9 real-world datasets in five domains confirm the superiority of the plugin models used. The authors presented the average percentage improvement of individual models and datasets, which is useful for reviewing the results.

•	The results with the proposed plugin are best for all datasets and backbone models used.

•	The prosed method requires only one core hyperparameter, the quantile q in the robust denormalizer, which is 0.75 by default.

•	The proposed method is flexible and can be used with any backbone model.

•	The authors conducted an ablation study, proving that all components are important. The authors explained how core components can be helpful data drift and concept drift mitigation.

•	The authors conducted computation time and memory usage study.

**Weaknesses:**

•	The overhead for lightweight models like DLinear is significant.

•	The authors did not provide average percentage increases in computation time and memory consumption for individual backbone models and datasets, which would be useful for quickly reviewing results.

•	Figure 2 suggests that statistics are calculated separately for historical and future mappings denormalization which is not true.

•	Lack of theoretical proof for the proposed plugin.

**Questions:**

•	Please refer to the weaknesses section.

•	Do the authors see a way to use the proposed method in datasets where timestamps are not available?

•	Do the authors plan to publish the exact same datasets used in the experiments?

•	It may be beneficial to see sample outputs from historical and future mappers.

**Limitations:**

As the authors mentioned, adding the plugin to backbone network cause additional computational time and memory usage.

---

> ### Author Rebuttal · Authors · 2024-08-05
>
> Thanks for your positive comments and insightful suggestions. Please find our response below.
>
> **Q1: Explanation of computation time and memory consumption.**
>
> As stated in Appendix D, higher prediction accuracy is typically accompanied by a higher computational cost. Balancing these needs requires consideration within the context of actual production demands. This study primarily focuses on enhancing prediction accuracy. In future work, we will consider model reduction to broaden the applicability of the proposed method. Additionally, we have not provided the percentage increase in computation time and memory consumption due to insufficient statistical significance. In the case of memory consumption, our proposed method increases consumption by 25MB for both DLinear (from 0.1MB) and TimesNet (from 150MB), representing percentage increases of 25000% and 17%, respectively. This statistical method may cause confusion and misunderstanding for readers.
>
> **Q2: Explanation of the misunderstanding caused by Figure 2.**
>
> We apologize for the confusion. As you noted, the statistics used in the inverse normalization for the historical mapping $\tilde{\mathbf{X}}$ and future mapping $\tilde{\mathbf{Y}}$ are indeed identical, as indicated in the parentheses in Equation 3. The two "Stats" are plotted in Figure 2 primarily for a more organized framework diagram.
>
> **Q3: Theoretical proof of the proposed method.**
>
> iTransformer does not utilize the timestamp information of the prediction window, which may contribute to its underutilization of timestamps. Therefore, we focus solely on the comparison between our proposed method and the summation scheme represented by Informer. The method proposed by Informer and our approach can be abstracted as early fusion (feature-level fusion) and late fusion (decision-level fusion). Early fusion (Informer) integrates modalities into a single representation at the input level and processes the fused representation through the model. Late fusion (GLAFF) allows each modality to run independently through its own model and fuses the outputs of each modality. Compared to early fusion, late fusion maximizes the processing effectiveness of each modality and is less susceptible to the noise of a single modality, resulting in greater robustness and reliability. This has been validated by extensive previous work [1, 2]. To mitigate the effect of noise and fully exploit the robustness of global information represented by timestamps, our proposed method adopts late fusion.
>
> **Q4: Applicability of the proposed method in the absence of timestamps.**
>
> Thanks to the timestep-level embedding and attention mechanism, our proposed approach can tolerate missing timestamps. Specifically, it only requires adding the corresponding mask matrix for the Attention-based Mapper to exclude the missing timestamps, allowing the entire framework to function seamlessly. We present the MSE results of the iTransformer backbone on two representative datasets. The history window is set to 96, and the prediction window to 192. The results in the table below clearly demonstrate the robustness of our proposed method to varying percentages of missing timestamps.
>
> |   Dataset   | iTransformer | + Ours 0% | + Ours 10% | + Ours 20% | + Ours 30% |
> | :---------: | :----------: | :-------: | :--------: | :--------: | :--------: |
> |   Traffic   |    0.3267    |  0.2909   |   0.3067   |   0.3225   |   0.3429   |
> | Electricity |    0.1674    |  0.1434   |   0.1490   |   0.1554   |   0.1691   |
>
> **Q5: Availability of the datasets.**
>
> The timestamp information we integrate has been extensively utilized, though ineffectively, by various existing methods. We introduce no additional information or assumptions. As detailed in Appendix A.1, all datasets employed (including timestamp information) are publicly accessible. We will also release the code and datasets upon completion of the paper review.
>
> **Q6: Visualization of the output from the mapper.**
>
> We visualize the Mapper and Denormalizer outputs for the two prediction cases in Section 4.3 of the paper **in Figure 1 of the Global Response attachment PDF**. From the experimental results, the Attention-based Mapper captures the majority of the shape information, while the Robust Denormalizer aligns the distribution of mapping values. Together, they provide comprehensive and robust assistance for accurate prediction of the backbone model.
>
> **References**
>
> [1] 2024, Foundations and Trends in Multimodal Machine Learning: Principles, Challenges, and Open Questions
>
> [2] 2023, A Comparative Analysis of Early and Late Fusion for the Multimodal Two-Class Problem

---

> > ### Comment · Reviewer_NzBw · 2024-08-11
> >
> > I appreciate the authors' rebuttal which clarified my concerns and convinced me to raise the overall rating.

---

> > > ### Author Response · Authors · 2024-08-11
> > >
> > > Thank you for thoroughly reviewing our rebuttal and deciding to raise the rating score. We appreciate your consideration and the time you have dedicated to evaluating our work.

---

### Author Rebuttal · Authors · 2024-08-05

We provide some images and tables **in the PDF attachment of the global response**, accompanied by detailed descriptions in the individual responses for each reviewer.

---

### Decision · Program_Chairs · 2024-09-25

**Decision:**

Accept (poster)

**Comment:**

This well-written paper has been assessed  by five knowledgeable reviewers who predominantly voted to have it accepted for NeurIPS (two full accept ratings, two weak acceptances, and one borderline rejection). The authors provided a comprehensive rebuttal to the initial criticism, and engaged in conversations with most of the reviewers. Those reviewers who communicated with the authors have their main concerns predominantly addressed. In summary, this work is viable for presentation at NeurIPS in spite of a more critical view of it by one of the five reviewers.